# Demographic causes and social consequences of adult sex ratio variation

Zitan Song [1,2] ✉, András Liker[3,4], Yang Liu [5], Robert P. Freckleton [6] & Tamás Székely[7,8,9] ✉

The adult sex ratio plays a crucial role in shaping breeding systems and traits linked to sexual selection. Recent studies associate adult sex ratio with mate choice, pair-bonding, and parenting, as the rarer sex gains advantages in mate selection and parental investment. However, the causal relationships between the demographic factors that generate adult sex ratio bias and its broader implications remain debated. Skewed adult sex ratios can result from sex-biased mortality and maturation, which influence mating and parental behaviours. Conversely, the costs of these behaviours may further drive sex differences in mortality and maturation, reinforcing adult sex ratio biases. Here, we compile demographic and behavioural data from 261 bird species across 69 families to examine these associations within a phylogenetic framework. Our analyses confirm that demographic traits are associated with adult sex ratio and reveal links between adult sex ratio, pre-copulatory sexual selection, and parenting. Phylogenetic path analyses further demonstrate that sex differences in mortality and maturation drive adult sex ratio biases, which subsequently influence mating and parenting rather than the reverse. This study provides a comprehensive analysis of the interplay between demography, social environment, and breeding systems, highlighting adult sex ratio as a crucial link. Our findings underscore the need for further research on the demographic underpinnings of social behaviour and reproductive strategies.

The adult sex ratio (ASR, typically expressed as the proportion of males in the adult population) serves as a cornerstone demographic trait that reveals a remarkable variation across natural populations[1]. Darwin[2] suspected that ASR is related to sexual selection: "*That some relation exists between polygamy and development of secondary sexual characters, appears nearly certain; and this supports the view that a numerical preponderance of males would be eminently favourable to the action of sexual selection.*" Recent studies also show that ASR plays a key role in the evolution of breeding systems, such as mate choice, pair bonding and parenting[1,3]. Theory predicts that in male-skewed populations, the males are expected to court more, have higher levels of extra-pair paternity and provide more care than in female-skewed populations[4–6]. Experimental, observational and comparative studies across a broad range of taxa that include fishes, reptiles, birds, mammals and humans are consistent with these predictions[1,7]. ASR can be influenced by various demographic factors, such as sex ratios at

[1]Co-Innovation Center for Sustainable Forestry in Southern China, College of Life Sciences, Nanjing Forestry University, Nanjing, China. [2]Comparative Socioecology Group, Department for the Ecology of Animal Societies, Max Planck Institute for Animal Behavior, Konstanz, Germany. [3]HUN-REN-PE Evolutionary Ecology Research Group, University of Pannonia, Veszprém, Hungary. [4]Behavioral Ecology Research Group, Center for Natural Sciences, University of Pannonia, Veszprém, Hungary. [5]State Key Laboratory for Biocontrol, School of Ecology, Sun Yat-sen University, Shenzhen, China. [6]School of Biosciences, University of Sheffield, Sheffield, UK. [7]Milner Centre for Evolution, Department of Life Sciences, University of Bath, Bath, UK. [8]HUN-REN-DE Reproductive Strategies Research Group, Department of Evolutionary Zoology and Human Behaviour, University of Debrecen, Debrecen, Hungary. [9]Department of Ethology, Eötvös Loránd University, Budapest, Hungary. ✉e-mail: songzitan@gmail.com; T.Szekely@bath.ac.uk

conception and at birth (BSR), sex differences in juvenile and adult mortality as well as differences in the age at sexual maturation. For instance, biased BSR can persist or even intensify into adulthood, leading to skewed ASR[8–10]. ASR can be further moulded by sex differences in mortality in juveniles and/or in adults[11–14], and males and females may exhibit sex differences in maturation ages, resulting in fewer adults of the slower-maturing sex in the population[10]. Therefore, a logical framework for understanding the key role of ASR in social evolution should consider several components of demography (i.e., BSR, sex differences in mortalities and maturation) as drivers of ASR variations. These variations, in turn may modulate breeding systems via the scarcity of one sex or the other[1,3,15,16] (Hypothesis 1, see Fig. 1a).

However, the causality between ASR and breeding systems is contested because an alternative logical framework suggests the association between the breeding systems and ASR could emerge via a different pathway wherein the costs of sexual competition (either pre-copulatory or post-copulatory) or parenting impose mortality costs which in tun, shifts ASR[1,7] (Hypothesis 2, Fig. 1b). Concomitantly, sexual selection involved in mate acquisition, courtship behaviour and competition can elevate mortality due to combat, rivalry, and mate search[17–19]; for instance, males with elaborate traits, or engaging in courtship or territorial defence often suffer from increased predation or heavy parasite load, leading to female-skewed ASRs[12,20,21]. Conversely, in species where females engage in more competition and fights, male-skewed ASRs may occur due to elevated female mortality[22]. The physiological costs of mate competition, such as immunosuppression from elevated testosterone, may also contribute to higher male mortality[23]. Additionally, competition for mates may require substantial time to acquire effective courtship strategies, to develop more pronounced ornaments, or to enhance fighting abilities – all of these could elevate the mortality of the more competitive sex even prior to maturation[10]. Lastly, in species that engage in parental care, the physiological or energetic demands and potential trade-offs with other life-sustaining activities could be greater for one sex than the other, potentially resulting in higher mortality of the caring sex that manifests as skewed ASR (i.e., the cost of reproduction[19,24]).

Here we use phylogenetic comparative analyses[25] to elucidate the evolutionary relationships among demographic factors, ASR, and breeding systems to assess the two major hypotheses of their associations (Fig. 1). Birds are excellent model organisms due to their substantial variation in ASR[3], the wealth of available data on their demography and breeding systems from extensive studies of natural populations[19], and their well-resolved phylogeny[26]. We have three objectives. First, to test the predictive power of the major demographic factors (i.e., BSR, juvenile mortality bias, maturation bias, and adult mortality bias) for interspecific variation in ASR. Second, we assess the influence of ASR on breeding systems by using three main components: pre-copulatory sexual selection (represented by sexual size dimorphism, plumage dimorphism and mating system), post-copulatory sexual selection (represented by relative testes mass and extra-pair paternity), and parental care. Finally, we use phylogenetic confirmatory path analysis[27] to evaluate the two competing pathways for understanding the associations among demography, ASR and breeding systems.

## Results

### Phylogenetic variation in ASR

ASR shows consistency within species. For 65 species, we collected ASR data from 2 to 7 populations, with a repeatability of 0.629 (95% CI: 0.514, 0.748, $p < 0.0001$). ASR, demographic variables and breeding systems show large interspecific variation (Fig. 2). For instance, the superorders Coraciimorphae (i.e., hornbills (Bucerotiformes), woodpeckers and toucans (Piciformes), kingfishers and rollers (Coraciiformes)) and Psittaciformes (i.e., parrots) exhibit strong male-biased ASR and typically experience female-biased mortality. In contrast, the order Strisores (i.e., hummingbird and allies) exhibits female-biased ASR, typically experiences male-biased mortality, and has a polygynous mating system where females provide more care for the offspring than males. Although ASR is generally male-biased across all birds (mean ± SE: $0.543 \pm 0.006$, $n = 261$ species), the order Passeriformes shows the largest interspecific variations across 113 species in ASR, mortality rates in both juveniles and adults, mating systems, and sexual plumage dimorphism.

### Demographic variables and adult sex ratios

Male-biased mortalities in both juveniles and adults, and male-biased maturation time predict female-biased ASR (Fig. 3, Table 1). Importantly, demography bias (i.e., the mean of the above three demographic variables) is a strong predictor of ASRs (Model 2, Table 1), which explained 38% of ASR variance (Table 1).

Birth sex ratios (BSRs) are unrelated to adult sex ratios (Supplementary Table 2), and we suggest two explanations. First, BSRs are typically close to 0.5 and less variable across species than ASR (variance$_{BSR}$ = 0.003; variance$_{ASR}$ = 0.008, $F$ test: $F = 0.389$, $p < 0.0001$, see Supplementary Fig. 1). Second, birth sex ratio has smaller sample size (103 species) than ASR (261 species). To deal with the latter issue, we boosted sample sizes for BSR by imputation (see Methods and Supplementary Table 3) and used the imputed iBSR data to test its

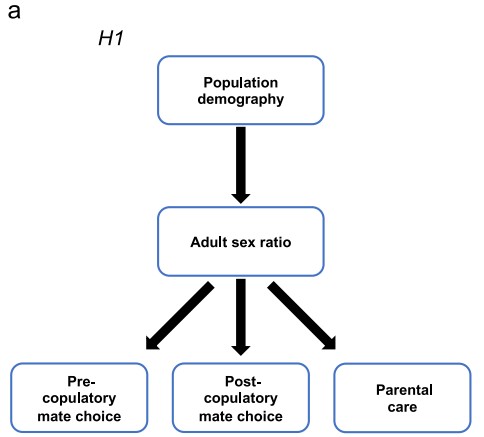

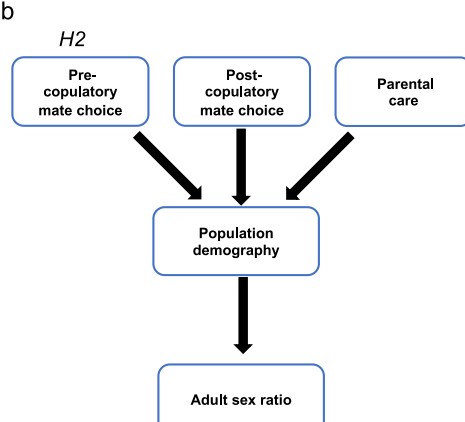

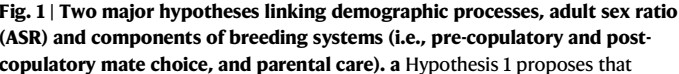

**Fig. 1 | Two major hypotheses linking demographic processes, adult sex ratio (ASR) and components of breeding systems (i.e., pre-copulatory and post-copulatory mate choice, and parental care). a** Hypothesis 1 proposes that demographic variables via ASR are the drivers of breeding system variation. Conversely, **b** Hypothesis 2 proposes that ASR emerges as the result of mortality costs of mate choice, sexual selection and parenting.

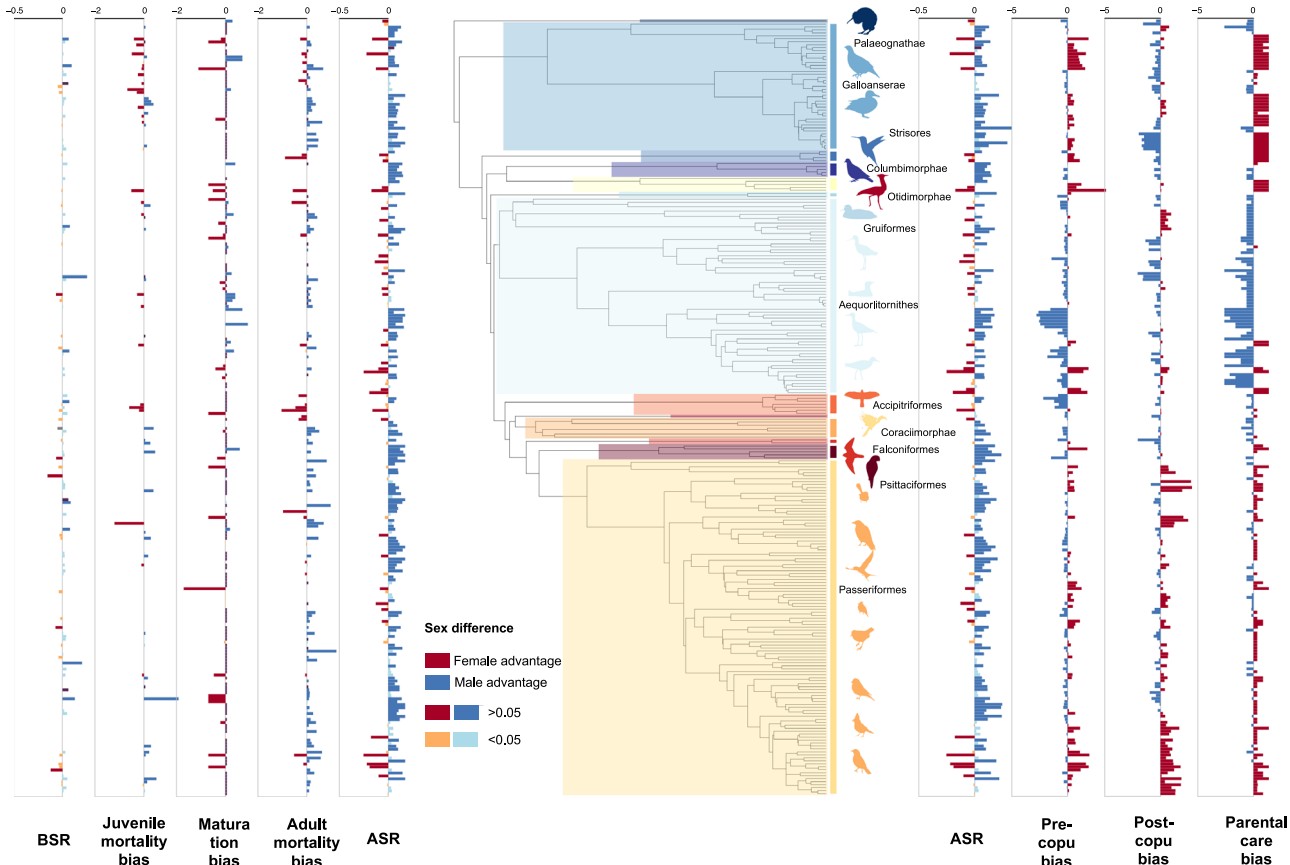

**Fig. 2 | Phylogenetic distribution of demographic variables (left), ASR and breeding system components (right) across 261 species from 69 avian families.** Colours indicate the direction of sex bias: red indicates female-biased, blue indicates male-biased. For sex ratio and adult mortality, colour intensity reflects the magnitude of the bias; orange and light blue mark values within ±0.05 of parity. For maturation bias, a short purple tick marks exactly zero difference between the sexes. Major avian clades follow the Prum et al.[27] backbone and are colour-coded on the phylogeny. Breeding systems are shown by the relative direction of pre-or post-copulatory bias and parental care (red indicates female-advantaged with female-biased care; blue indicates male-advantaged and male-biased care). Bird silhouettes were obtained from PhyloPic (http://phylopic.org). The silhouette of *Recurvirostra americana* is licensed under Attribution 3.0 Unported (CC BY 3.0) by Alexandre Vong; the remaining silhouettes are released under CC0 1.0 or Public Domain Mark 1.0.

association with ASR. Nevertheless, iBSR and ASR was not associated (Supplementary Table 4), although note that the iBSR had low reliability (Supplementary Table 3). Therefore, due to the lack of association between BSR and ASR, and the low reliability of imputed BSR, we did not consider BSR in subsequent analyses.

## Adult sex ratios and breeding systems

We imputed relative testes mass (iTestes) and extra-pair paternity (iEPP) to increase sample sizes (Supplementary Table 3), the imputed iTestes and iEPP were used to represent post-copulatory bias (iPCB) in further test of their associations with ASR. ASR is associated with pre-copulatory bias since female-skewed ASRs predicted male-bias in mating systems, body size and plumage dimorphism (Fig. 4, Table 2 and Supplementary Table 5), and ASR also predicted parental care bias (Fig. 4, Supplementary Table 5). However, neither the imputed post-copulatory bias (iPCB) nor the raw data of relative testes mass and extra-pair paternity were associated with ASR (Table 2 and Supplementary Table 5).

In multi-predictor models, pre-copulatory bias remained the only significant predictors of ASR suggesting that pre-copulatory bias has a key role driving associations between ASR and breeding systems (Table 2). An analysis of the variance inflation factor (VIF) indicates that there is no significant multi-collinearity in the multi-predictor models when considering pre-copulatory bias, post-copulatory bias, and parental bias (Supplementary Table 6).

## Phylogenetic path analysis

To test the two main hypotheses among demographic processes, ASR, and breeding systems (Fig. 1), we conducted phylogenetic path analyses under two complementary frameworks that differ in how phylogenetic covariance is parameterised. First, following the approach proposed by Santos[27], we implemented a trait-level transformation framework in which phylogenetic structure is accounted for prior to path estimation (Fig. 5 and Supplementary Figs. 2 and 3). In the first model set, we used imputed data for post-copulatory bias (iPCB) since the number of species available for post-copulatory biases would have limited the analyses to 25 species. Using the imputed dataset for post-copulatory biases (iPCB), the sample size increased to 65 species. Based on the two main hypotheses, we constructed six path models (see Supplementary Fig. 2). This path analysis yielded two best-fitting models that were not rejected by the d-separation test (Fisher's C) and had the lowest AICc values (delta AICc < 2): both best models supported Hypothesis 1 (Fig. 5a, see also Supplementary Table 7 and Supplementary Fig. 4a).

In the second model set, since the post-copulatory bias is not associated with ASR (Table 2 and Supplementary Table 5), we only included pre-copulatory bias and parental care bias in the path model using only the non-imputed data (*n* = 67 species). We constructed six path models following the two main hypotheses, consistently with the above rationale (see Supplementary Fig. 3). The latter analyses

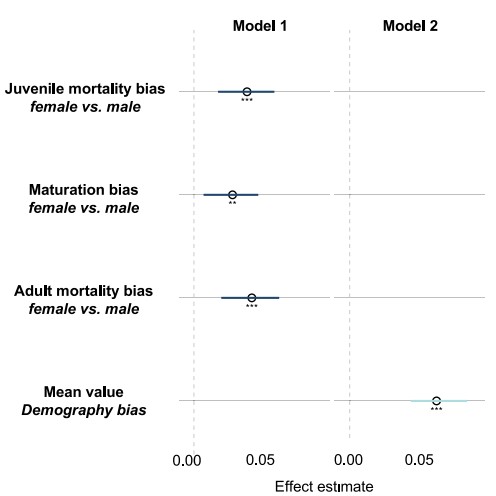

**Fig. 3 | Demographic variables in relation to adult sex ratio (dependent variable) in birds (see Table 1).** Model 1 includes sex bias (female relative to male) in juvenile mortality, adult mortality and maturation. Model 2 includes demographic bias (i.e., the mean bias of juvenile mortality, adult mortality and age at maturation). Points show posterior mean estimates and bars indicate 95% posterior credibility intervals; estimates are based on 67 species. Significance codes indicate MCMC-derived two-sided *P* values (\**P* < 0.05, \*\**P* < 0.01, \*\*\**P* < 0.001).

**Table 1 | Adult sex ratio (ASR) in relation to demographic variables in phylogenetically controlled MCMC models in birds**

|  | Posterior mean | Lower–upper 95% CI | pMCMC |
|---|---|---|---|
| *Model 1* | | | |
| Intercept | 0.531 | 0.506–0.555 | 0.0006 |
| Juvenile mortality bias | 0.033 | 0.015–0.050 | 0.0006 |
| Maturation bias | 0.024 | 0.006–0.040 | 0.009 |
| Adult mortality bias | 0.036 | 0.017–0.053 | 0.0006 |
| *Model 2* | | | |
| Intercept | 0.531 | 0.505–0.553 | 0.0006 |
| Demography bias | 0.054 | 0.038–0.073 | 0.0006 |

Model 1 examines ASR in relation to juvenile and adult mortality bias, and maturation bias. Model 2 examines ASR in relation to demography bias, i.e., the mean of juvenile and adult mortality biases, and maturation bias. Estimates are presented as posterior means with 95% posterior credibility interval (CI); pMCMC denotes the MCMC-derived two-sided P value (*n* = 67 species with data for all demographic variables). All biases were calculated as female value minus male value.

provided two best fitting models (see Supplementary Table 8, and Supplementary Fig. 4b), and both best fitting models were consistent with Hypothesis 1 (Fig. 5b). Within the transformation framework, all four best models showed consistent fit statistics (Supplementary Table 7 and 8), as indicated by Tucker–Lewis index (TLI > 0.95), Bentler's comparative fit index (CFI > 0.95), root mean square error of approximation (RMSEA < 0.06), and standardised root mean square residual (SRMR < 0.08)[28,29].

In parallel, we implemented a residual-based phylogenetic path analysis using the phylopath (version 1.3.1)[30] R package, in which phylogenetic covariance is modelled conditionally within each regression component (see the 'Methods' for details). This approach produced concordant top-ranked models (Supplementary Table 9) across all model sets, supporting the same causal structure identified under the transformation framework.

Together, the convergence of results across frameworks supports the proposition that demographic bias shapes ASR, which subsequently impacts the breeding systems. These results also indicate that ASR has either a direct effect on pre-copulatory bias, which in turn influences parental care, or ASR directly feeds into parental care, which then may have a knock-on effect on pre-copulatory bias.

## Discussion

Based on a broad-scale phylogenetic analysis of interspecific variation in ASR, we identified three primary findings. Firstly, we established sex different mortalities and maturation as the main drivers of ASR variation. Secondly, we showed that ASR predicts both pre-copulatory sexual selection and parental care bias, but not post-copulatory bias. Thirdly, confirmatory path analyses consistently supported the demography → ASR → breeding system hypotheses in contrast to the breeding system → demography → ASR hypotheses. Together, these results have the potential to advance studies of sexual differences, sex roles and breeding systems by supporting the key role of ASR in social evolution.

Our first finding confirms the impact of demographic variables on ASR. Species with higher male mortality rates during the juvenile and/or adult stage exhibit a pronounced female-biased ASR. ASR data, although collected from multi-populations among different years, show remarkable stability among populations (repeatability: 0.629). Moreover, other demographic variables were also estimated in different populations and from different time periods from ASRs, and often using a variety of methodologies in the field (see 'Methods'), which make those associations are remarkable.

Interestingly, we found no detectable impact of the birth sex ratio (BSR) on adult sex ratio (ASR), also see refs. 31,32. Across the 103 species included in our dataset, BSRs clustered closely around parity, in accordance with the Fisher's principle[33], which predicts equal investment in male and female offspring. The parity in hatching sex ratio aligns with expectations for species with heterogametic chromosomal sex determination[34]. However, we acknowledge that BSR data were available only for a limited number of species and often derived from small sample sizes. Therefore, our analysis cannot rule out the possibility that subtle variation in BSR contributes to ASR variation in some cases. Nevertheless, the relative low variance in BSR compared to high variance in ASR across species supports the view that post-birth factors, rather than birth ratios, play a more significant role in determining ASR. This finding aligns with previous studies, which highlighted the role of postnatal factors like differential mortalities and maturation in influencing ASR[10,14,32].

Adult sex ratio reflects the relative numbers of adult males and females and is determined by two components: recruitment to adulthood and the time spent in the adult stage. The latter depends primarily on annual adult survival and adult lifespan. However, the data available on sex-biased lifespan are currently limited and heterogeneous in wild birds and can be heavily influenced by a small cohort of long-lived individuals, which is not temporally aligned with the age classes to which ASR is most sensitive (young and prime breeders). In a supplementary analysis (*n* = 16 species, Supplementary Table 10), sex-biased lifespan showed no detectable association with ASR, whereas sex-biased adult mortality remained a strong predictor. Based on limited reliable data on sex-biased longevity, we suggest that survival differences, rather than longevity per se, largely drive ASR variation. Progress toward more reliable data will require cohort-based long-term monitoring, harmonised lifespan/adult-tenure definitions, and empirically validated capture–mark–recapture models with detection corrections.

Beyond our core demographic predictors, there are other potential recruitment variables like migration and sex-biased dispersal that could modulate ASR variation. In principle, protandry and sex differences in routes, distances, or winter niches may expose males

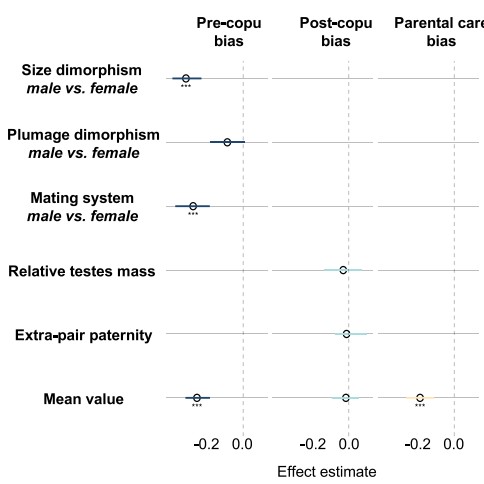

**Fig. 4 | Associations between ASR (response variable), pre-copulatory bias, post-copulatory bias (iPCB), and parental care bias.** Points show posterior mean estimates and bars indicate 95% posterior credibility intervals; sample sizes vary among analyses (see Supplementary Table 5). Significance codes indicate MCMC-derived two-sided P values (*$P < 0.05$, **$P < 0.01$, ***$P < 0.001$).

and females to distinct energetic demands and hazards, potentially generating sex-biased adult survival and thereby shifting ASR[35,36]. However, supplementary analyses using our large ASR database, found no detectable association between migration status/distance and ASR, nor evidence that migration aligns with sex-biased mortality (Supplementary Table 11). One parsimonious explanation is that, for many species, both sexes traverse broadly similar migratory circuits within a season, which may equalise risks. Likewise, sex-biased dispersal could alter local ASR if emigration by one sex is not balanced by immigration, and the dispersing sex may incur higher mortality[37,38]. In a preliminary test with 63 species (Supplementary Table 12), dispersal bias nonetheless showed no effect on ASR or on sex-biased mortality. This lack of association is consistent with demographic compensation: populations that lose one sex to emigration are replenished by immigrants of the same sex. Additionally, in a study test more than 200 bird species also indicate minimal sex differences in dispersal distance, further limiting the expected impact on ASR[39]. Overall, while migration and dispersal remain plausible contributors in some systems, our results suggest that their influence is limited at broad macroecological scales.

The traditional explanation for the imbalance in parental care between males and females has been attributed to the difference in gamete size (anisogamy), as elucidated by Robert Trivers[40], who posited that females, having invested more in pre-fertilisation, are inclined to invest more in post-fertilisation parental care than males. However, there has been increasing criticism of this argument, with proponents contending that optimal decision-making should be predicted by future costs and benefits rather than past ones[5,6,41]. Moreover, an individual's future reproductive success is highly dependent on the ASR, as males cannot, on average, reproduce faster than females if the ASR is balanced[42]. In our study, while ASR initially showed a significant correlation with parental care bias (Supplementary Table 5), this relationship disappeared when pre- and post-copulatory mate choice were included in the model (Table 2). Nevertheless, our phylogenetic path analyses revealed that ASR influences pre-copulatory sexual selection, which is in turn associated with parental care bias. This suggests that biased parental care may not be directly shaped by potential mate imbalance, but rather co-evolves with sex-specific mating strategies under sexual selection. Importantly, variance inflation factor (Supplementary Table 6) analyses showed no problematic multicollinearity, indicating that this pattern is not driven by statistical suppression. These findings are consistent with recent theoretical studies suggesting that parental care evolves in concert with sexual selection, sex-

biased mortality, and maturation, rather than being a direct consequence of ASR[5,6]. Together, our results suggest that biased parental care may be a by-product or may co-evolve with sexual selection, rather than being directly influenced by ASR.

There is an implicit assumption that an optimal evolutionary response to competition is to invest more heavily in traits that enhance competitiveness. However, the reproductive success of abundant males could be low when there are few females available for mating[41]. In fact, a meta-analysis indicates that with increasing bias in ASR, male courtship rates may decrease and instead, mate guarding tends to intensify[43]. Moreover, the less common sex, benefiting from increased opportunities to mate with multiple partners through extra-pair copulations and polygamous relationships, may also exhibit heightened sexual dimorphism due to selective pressures favouring traits that maximise these mating opportunities[44]. Our findings suggest that a male-skewed ASR is associated with female-biased sexual dimorphism, resulting in larger and more ornamented females. Additionally, our results indicate a negative correlation between ASR and mating system dynamics, implying that a male-biased ASR may promote polyandry. A male-skewed ASR may enhance female fertilisation success, suggesting stronger sexual selection pressures on females[45], which could account for the observed trend towards larger size and more elaborate features in females. These scenarios suggest that competition does not necessarily intensify among the more prevalent sex, but instead enhances the reproductive benefits for the less common sex, as they have the opportunity to access multiple sexual partners[1,44].

Interestingly, we found no correlation between ASR, and relative testes mass or the frequencies of extra-pair broods (Fig. 4, Supplementary Table 5). Previous works show that the frequency of extra-pair broods was higher in male-biased ASR in monogamous species, but no clear pattern for all species[46]. Post-copulatory mate choice represents a strategy to enhance reproductive success without incurring the costs associated with parental care, often linked to strong sexual selection during pre-copulatory mate choice[47] and a reduction in male investment in parental care[48]. Despite these dynamic associations, extra-pair mating behaviour can be costly for females due to the effort required to seek extra-pair mates[49], decreased investment from their social partner[50], and the risk of sexually transmitted disease[51]. The primary benefit for females engaging in extra-pair mating may lie in accessing sperm from multiple males, serving as a hedge against infertility[52]. Moreover, the impact of extra-pair mating may simply manifest as a tendency towards polygamy in monogamous species, and its association with ASR could be obscured by the social mating system[46].

Our phylogenetic path analyses support the hypothesis (Fig. 1a) that demographic biases result in biased ASRs, which subsequently exerts a significant influence on breeding system. Indeed, these results align with previous theoretical studies, demonstrating that a male-biased ASR is associated with a higher proportion of male care only when ASR varies due to a general change in male mortality[5]. A male-biased ASR, resulting from lower male mortality, makes it cheaper for males to provide care[5]. Moreover, decreased male care likely contributes to increased sexual dimorphism. This suggests that the benefits of reduced parental investment, aimed at exploiting advantages from sexual conflict and encouraging greater investment from mates, may not be sustainable even in environments with abundant potential partners[5,6,41].

Contrary to previous research[10], our path analysis suggests that maturation biases may drive, rather than result from, the ASR variation through sexual selection. This indicates that the larger sex requires more time to mature. However, our path analysis shows that maturation biases, as one of the demographic variables, do not directly affect sexual size dimorphism or plumage dimorphism (pre-copulatory mate choice). Instead, the influence of demographic variables on pre-copulatory mate choice is mediated by ASR since the late-maturing sex

**Table 2 | Adult sex ratio (response variable) in relation to pre-copulatory bias, post-copulatory bias (imputed, iPCB), and parental care bias in phylogenetically controlled MCMC models**

|  | Posterior mean | Lower – upper 95% CI | pMCMC |
|---|---|---|---|
| Intercept | −0.186 | −0.610–0.254 | 0.380 |
| Pre-copulatory bias | −0.543 | −0.704–0.374 | 0.0006 |
| Post-copulatory bias (iPCB) | 0.052 | −0.084–0.196 | 0.454 |
| Parental care bias | 0.019 | −0.166–0.206 | 0.877 |

Pre-copulatory bias is defined as the mean of size dimorphism, plumage dimorphism and social mating system bias, whereas post-copulatory bias is defined as the mean of relative testes mass and extra-pair paternity. Estimates are presented as posterior means with 95% posterior credibility intervals (CI); pMCMC denotes the MCMC-derived two-sided P value (n = 223 species). All bias were calculated as female value minus male value.

is underrepresented in the population due to the longer time it takes to mature[3]. Nevertheless, demographic variables may exert evolutionary feedback on mate choice and vice versa, the relationships that phylogenetic path analyses cannot test at present, as it is based on directed acyclic graphs and thus excludes reciprocal causation. While path analyses, unlike experiments, cannot establish causality, they excel at comparing alternative scenarios that illustrate different causal relationships between variables[53]. In this context, the alternative causal directions tested among demographic traits, ASR, and mating systems in our study are appropriate and interpretable. The discrepancy with previous findings[10] may arise from our use of an integrative approach, treating population demography as a composite variable (including maturation bias, adult and juvenile mortality bias), and accounting for both pre- and post-copulatory sexual selection in a unified framework.

Another variable that may influence the relationship between demography, ASR, and breeding systems is group living. Group living can create economic conditions that allow for the potential monopolisation of multiple mates, which increases mate competition and results in more skewed polygamy[54]. Species that live in groups may benefit from reduced predation risk[55] or a buffer against fluctuating environmental conditions[56], which can have different effects on males and females due to their differing responses to social conflict. Additionally, the clumped distribution of food and habitat may facilitate group living and increase group size[54]. However, the potential ecological effects of group living on the relationship between demography, ASR, and breeding systems still need to be fully understood.

The causal relationships between demography, ASR, and breeding systems may vary among different vertebrate classes. For example, in mammals male–male competition for mates appears to be more intense than in birds[57], potentially leading to stronger effects on sex-bias mortality[58]. Mammals typically exhibit predominant polygamy compared to birds[59], and the disparity in parental care between males and females is more pronounced in mammals due to exclusive female gestation and lactation. This suggests a higher energy demand and mortality implications of pregnancy and lactation for females could further impact survival[60]. In fact, mammalian populations often show a female-biased ASR, in contrast to the male-biased ASR commonly seen in birds[61,62]. However, the differences in the causal relationships between demography, ASR, and breeding systems in mammals and birds remain a possible area for further studies.

In conclusion, our study supports the pivotal role of the adult sex ratio in connecting demographic factors with breeding systems across a broad range of avian families. We have demonstrated that demographic biases—such as juvenile and adult mortality rates and sex-specific maturation times—significantly influence ASR, which in turn impacts breeding systems. Our results reveal that ASR predominantly affects pre-copulatory mate choice, while its effects on post-copulatory mating behaviours and parental care are less pronounced. Our research thus highlights the complexity of ASR and its critical role in linking demographic factors with breeding systems, offering valuable insights into the evolution of these elements. Future analyses should investigate

these hypotheses in other well-studied taxa for instance mammals, frogs and beetles, in a phylogenetic framework. Future investigations in the field or in the laboratory can further scrutinise the hypothetical relationships we reported here using different organisms and different methodologies. Taken together, the fundamental idea connecting sexual selection to sex ratios, as proposed by Charles Darwin[2] requires comprehensive development and empirical validation in future studies.

## Methods
### Data collection
In this study, we perform comparative analyses by gathering data on sex ratio, population demography, and breeding system from various published resources.

### Adult sex ratio
The adult sex ratio (ASR) was defined as the proportion of males in the adult bird population within the study area for each species. Our initial dataset was based on a previous study[44] which included 185 bird species. We further compiled additional published data to expand the dataset. To update our database beyond 2019, we conducted a search through the Web of Science and Google Scholar using the keywords 'birds' and 'sex ratio.' After reviewing 172 publications, we were able to gather ASR data for 42 additional species. The final dataset included 261 species. In cases where multiple ASR values were available for a species, we calculated the mean value. We found that ASR estimates were consistent across different populations of the same species, as indicated by a high repeatability (Repeatability = 0.629, with 191 populations of 65 species) estimated using the rptR (version 0.9.23)[63] package in R version 4.1.1. Moreover, sample size did not appear to influence ASR estimates, as reported by previous studies[14].

### Demography bias
**Birth sex ratio.** We collected birth sex ratio (BSR) data, which was defined as the proportion of males in a brood, for species with available ASR information from existing comparative datasets[14]. For species without available data, we conducted a literature search using the search engines Web of Science and Google Scholar. We used the search terms 'species name' and 'sex ratio' along with keywords such as 'offspring,' 'fledging,' 'hatching,' 'neonate,' or 'juvenile' to identify relevant literature. We gathered both fledging sex ratio and hatching sex ratio data from the literature search. In total, we collected hatching sex ratio data for 71 species and fledging sex ratio data for 63 species. As there was no significant difference was found within species between hatching and fledging sex ratios (paired t test: t = −0.445, n = 31 species, p-value = 0.657), we combined the two into a single measure of offspring sex ratio. Given that fledging sex ratios account for sex-specific nestling mortality, they provide a more accurate estimate of the sex composition entering the juvenile stage and thus are more relevant to adult sex ratio (ASR). Therefore, we used fledging sex ratios as the primary source (n = 63) and supplemented them with hatching sex ratios when fledging data were unavailable (n = 40). This

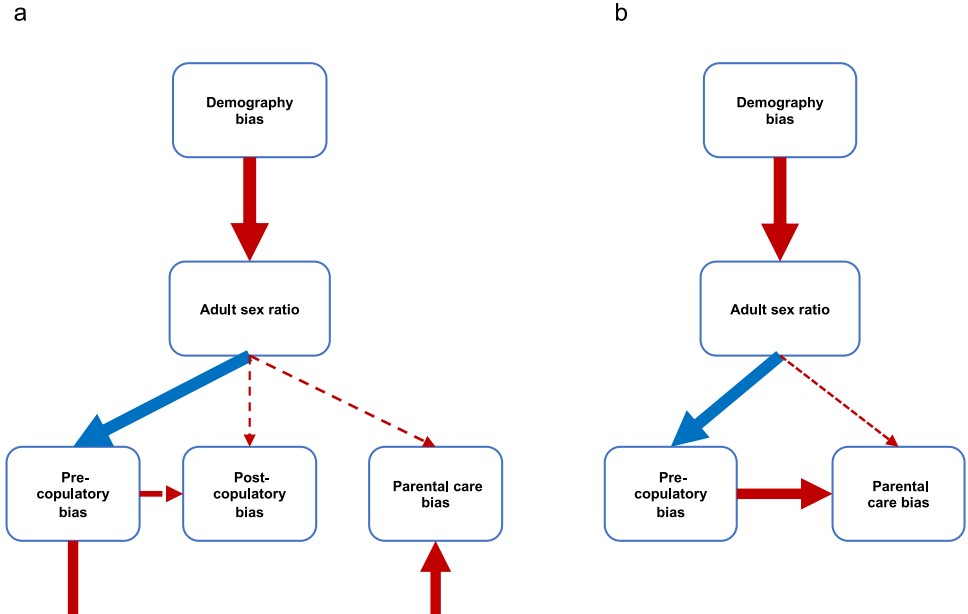

**Fig. 5 | Phylogenetic confirmatory path analyses with and without post-copulatory bias. a** Analyses including post-copulatory bias based on imputed data ($n = 65$ species), and **b** analyses excluding post-copulatory bias and based on non-imputed data ($n = 67$ species). Significant associations are shown by thick arrows whereas dotted arrows indicate non-significant associations. Red lines indicate positive relationships (for instance, female take longer to mature and/or suffer higher mortality leading to male-skewed ASR), and blue arrows indicate negative relationships (e.g., male-skewed ASR leading to female-biased sexual dimorphism and polyandrous mating system).

approach allowed us to increase the sample size and improve the statistical power of our analysis.

**Mortality bias.** We collected annual mortality rates from studies that estimated mortality (or survival) separately for males and females. Mortality rates were classified into two age classes: juvenile (before the age of first reproduction) and adult (after first reproduction). Most of the adult mortality data were obtained from the work of ref. 44, while the additional data for missing species were obtained from literature searches by using the keywords 'species name', 'survival', and 'mortality' in Web of Science and Google Scholar. Mortality rates were always collected from the same population for both sexes. In total, we obtained juvenile mortality rates for both males and females in 68 species, and adult mortality rates in 150 species. For juvenile mortality, and data were typically available from a single population per species. In contrast, adult mortality rates were available for multiple populations in some species, and in those cases, we used the species mean. Adult mortality showed high repeatability across populations (Repeatability: females = 0.678, males = 0.733, estimated using the R package rprR[63]), indicating that the species-level estimates are robust (based on 73 records from 35 species). We calculated sex differences in juvenile and adult mortality rates as the degree of female-biased mortality, expressed as the log ratio of female to male mortality (i.e., log (female mortality/ male mortality)), also referred to as sex-specific mortality.

**Maturation bias.** We collected data on the age of sexual maturation for males and females from available comparative databases[10], and supplemented from Web of Science and Google Scholar. In total, we gathered information on the age of sexual maturation (expressed in months) for 222 species. We calculated the maturation bias as the ratio of female age of maturation to male age of maturation (i.e., log (female age of maturation/male age of maturation)).

**Pre-copulatory bias**
**Sexual size dimorphism.** We collected data on male and female body mass for 254 species from various available databases (see

Supplementary Data for details). We calculated sexual size dimorphism as log (male mass/female mass).

**Sexual plumage dimorphism.** Sexual plumage dimorphism was estimated for 261 species based on data from ref. 64. This value of plumage dimorphism estimated from five body regions (head, back, belly, wings and tail) using the following scheme: −2, the female was substantially brighter and/or more patterned than the male; −1, the female was brighter and/or more patterned than the male; 0, there was no sex difference in the body region, or there was a difference but neither could be considered brighter than the other; 1, the male was brighter and/or more patterned than the female; 2, the male was substantially brighter and/or more patterned than the female. Since values among the five body regions were correlated, we used principal component analysis with varimax rotation using the principal() function from the psych[65], reducing them to one component (Supplementary Table 1a).

**Mating system.** To collect data on the social mating system, we followed the 0–4 points scale developed by ref. 66, which scores the incidence of social polygamy for each sex. A score of '0' indicates no or very rare polygamy (<0.1% of individuals), '1' indicates rare polygamy (0.1–1%), '2' indicates uncommon polygamy (1–5%), '3' indicates moderate polygamy (5–20%), and '4' indicates common polygamy (>20%). In total, we collected social polygamy data for both males and females across 251 species and calculated the mating system bias as the difference between the male and female scores. This classification reflects social, rather than genetic, mating systems (e.g., those inferred from extra-pair copulations) and is therefore more relevant to parental investment and reproductive behaviour, aligning better with our framework and with our previous studies[19,64].

**Post-copulatory bias**
**Testes mass and extra-pair brood.** Testes mass was collected for 164 species from ref. 67. To control for morphology-to-body size allometries, we calculated residual testes mass by regressing log10-

transformed testes mass against log10-transformed male body mass. Extra-pair brood data for 109 species were collected from ref. 68.

## Parental care bias

We measure sex differences in parental care by comparing female participation to male participation in three behaviours: incubation and chick feeding and chick defence, across 252 species. Female participation was scored on a 5-point scale (0: no female care, 1: 1–33% female care, 2: 34–66% female care, 3: 67–99% female care, 4: 100% female care; see[66] for a similar approach). Since precocial birds do not feed their young, we combined female feeding participation for altricial and semi-precocial species with female chick defence participation for pure precocial species and referred to this combined behaviour as 'chick attendance'. As female chick attendance value is correlated with female brooding value, we conducted a principal component analysis using the varimax rotation method in the R package psych (version 2.5.6)[65] to reduce these two parental care behaviours to one component (see Supplementary Table 1b).

## Data imputation

Given the presence of incomplete data sets of testes mass and extra-pair paternity, we conducted data imputation following[69,70]. Among species used in the imputation, raw data was available for 66% of testes mass and 44% of extra-pair paternity. To increase the robustness, we imputed variable with control variable (see, Supplementary Table 3), including: body mass, development mode (precocial, semi- precocial, semi-altricial and altricial), development period (sum of incubation and fledging periods), clutch size, male plumage colour, and local precipitation.

Multiple imputation was performed using a Brownian model of trait evolution and including Pagel's lambda to allow for varying amounts of phylogenetic dependence. We used the Rphylopars (version 0.3.10)[71] package in R version 4.1.1. A leave-one-out cross-validation was performed to test the accuracy of the imputation by deleting the non-missing data points one at a time, re-running the imputation, and comparing the imputed values with the original observations. This process was conducted for every observed data point in the dataset, allowing us to evaluate imputation accuracy across the full dataset. The results of the correlation are shown in Supplementary Table 3.

We also attempt to impute the birth sex ratio, which was available for 44% of the raw data, but this yielded low reliability (Supplementary Table 3), so we didn't include imputed BSR in the analysis.

## Statistical analyses

All statistical analyses were carried out in the R 4.1.1 environment[72]. We began by fitting a phylogenetically controlled mixed model using MCMCglmm (version 2.36)[25] package in R to assess the effects of demographic bias variables on adult sex ratio (ASR). These variables included birth sex ratio, juvenile mortality bias, maturation bias, and adult mortality bias, with the goal of identifying which components drive ASR. Due to the limited effect of birth sex ratio on ASR (Supplementary Table 2), we excluded it from following analyses. To assess the overall effect of demography bias on ASR, we standardised values of juvenile mortality bias, maturation bias, and adult mortality bias to a common scale by centreing and scaling each variable, and then averaged them to derive a composite demography bias index. This approach assumes that all three biases represent distinct but complementary components of sex-biased survival processes across life stages. Averaging their standardised values provides an integrative measure of overall demographic asymmetry influencing adult sex ratio. To highlight any potential biases resulting from data imputation, we additionally applied the model to raw data without any imputed values, encompassing a total of 67 species.

Subsequently, we employed phylogenetically controlled mixed models to examine the effects of ASR on both pre- and post-copulatory bias, and parental care bias. To ensure that potential multicollinearity among predictors did not obscure the individual effects—particularly for breeding system variables—we calculated variance inflation factors (VIFs) for all models. Pre-copulatory bias was calculated as the average of standardised values for sexual size dimorphism, sexual plumage dimorphism, and mating system. For sexual plumage dimorphism, we applied a Principal Component Analysis (PCA) to reduce the dimensionality of the plumage dimorphism values from five body regions into a single composite value (PC1; Supplementary Table 1). For post-copulatory bias, we used relative testes mass as an indicator of male physiological investment in sperm competition, and extra-pair paternity as the realised outcome of post-copulatory mating processes, encompassing both male–male sperm competition and female choice. These two measures capture distinct but complementary aspects of post-copulatory sexual selection, and together provide a more integrative, sex-inclusive representation of post-copulatory bias. Similarly, post-copulatory bias was quantified by averaging standardised values of relative testes mass and extra-pair paternity. Overall, we adopted this averaging approach under the assumption that traits within each component contribute equally, allowing us to construct unified indices of pre-copulatory, post-copulatory, parental bias and demographic bias that can be directly compared in their associations with ASR. In contrast, for parental care bias, we used the first principal component (PC1) from a PCA of female participation in brooding and chick attending, as these variables were measured on the same scale and could be meaningfully combined. PC1 captured the major axis of variation in female parental effort and was used as an index of overall female participation in breeding. In addition, we modelled bi-variable relationships between ASR and each breeding system index, conducting models for both imputed and raw data.

All models were fitted using an MCMC-based mixed modelling approach using the package MCMCglmm[25]. We used chains of 75,000 iterations, with the first 7500 iterations discarded as burn-ins and thinned every 40 iterations. Inspection of the final MCMC samples did not show any sign of autocorrelation. For all models, inverse-gamma priors were used for residual variances (parametrised as inverse-Wishart with $V = 1$ and $v = 0.002$). The prior for phylogenetic effect was formed as a weakly informative half-Cauchy density (parameter-expanded priors with $V = 1$, $v = 1$, $\alpha\mu = 0$ and $\alpha V = 10,000$). Priors for fixed effects were left as default (Gaussian densities with $\mu = 0$ and large variance).

## Phylogenetic path analysis

To assess the two hypotheses depicted in Fig. 1, we used phylogenetic path analyses to compare the two sets of path models corresponding to different hypotheses for the relationships linking ASR, breeding system and demography bias. Although path analyses—unlike experiments—cannot infer causality, it is a suitable method to compare alternative scenarios representing different causal relationships between variables[53].

Model 1 assumes that demography bias influences ASR, which in turn influences breeding system (Fig. 1a, H1). Demographic bias is represented by the average of standardised values for juvenile mortality bias, maturation bias, and adult mortality bias. The breeding system is characterised by three variables: pre-copulatory bias (the average of standardised values for sexual size dimorphism, plumage dimorphism, and mating system), post-copulatory bias (the average of standardised values for relative testes mass and extra-pair paternity), and parental care bias. Three variants of this model were examined: Model 1a, which assumes no correlations among the three breeding system variables; and Models 1b and 1c, which explore relationships among breeding system variables (refer to Supplementary Fig. 2). Model 2 hypothesises that the breeding system influences

demographic bias, which in turn affects ASR (Fig. 1b, H2), effectively mirroring Model 1 (as shown in Supplementary Fig. 2). Due to the lack of an association between post-copulatory bias and ASR (Table 2 and Supplementary Table 5), as well as limited data on post-copulatory bias, we conducted path analyses without including post-copulatory bias, using only non-imputed data ($n = 67$ species, Supplementary Fig. 3).

We used two complementary approaches to fit phylogenetic path models to the data. First, we followed the method proposed by ref. 27 and pre-transformed each variable using its own maximum-likelihood estimate of Pagel's $\lambda$, then fitted the path model with standard SEM tools. This approach accounts for phylogenetic structure at the trait level prior to model specification and yields phylogenetically independent contrasts for each variable. However, because $\lambda$ is estimated separately for each trait before path estimation, phylogenetic covariance is accommodated at the variable level rather than conditionally within each regression component. This process involved: (1) independently estimating $\lambda$ for each variable using maximum likelihood; (2) applying these variable-specific $\lambda$ values to recalibrate the phylogenetic tree into a unit tree; and (3) utilising the transformed tree to compute phylogenetically independent contrasts for each variable, achieved through the pic() function in the R package ape (version 5.8.1)[73]. This procedure was replicated for every variable, and the derived phylogenetically transformed values were then employed in fitting the path models. In the analysis second phase, we assessed the model fit utilising the d-separation method, as delineated by ref. 53 and implemented in the R package piecewiseSEM (version 2.3.0)[74].

Candidate models that were not rejected by the d-separation test (Fisher's C) were compared using Akaike information criterion (AICc), and models with ΔAICc <2 were considered to have substantial support. Additionally, we measured the model fit of individual models using four of the most widely used indexes by the R package lavaan (version 0.6.20)[75]: Tucker-Lewis index (TLI), Bentler's comparative fit index (CFI), root mean square error of approximation (RMSEA), and standardised root mean square residual (SRMR). TLI and CFI > 0.95, RMSEA < 0.06, and SRMR < 0.08 are typically interpreted as indicating acceptable relative fit[29]. To assess the impact of phylogenetic uncertainty on our results, we replicated the phylogenetic path analyses following[27] across 100 phylogenetic trees randomly from the posterior distribution published by ref. 76. By employing this approach, we aimed to identify the most parsimonious and well-supported models, allowing us to make robust inferences about the causal pathways underlying the combined effects of demography and reproductive social behaviour on ASR.

Secondly, we repeated the analyses using the residual-based method developed by ref. 77, implemented in the R package phylopath[30]. In this framework, phylogenetic covariance is modelled conditionally within each regression component by estimating $\lambda$ in the residual error structure of each equation. This formulation is consistent with conventional PGLS approaches, in which $\lambda$ is estimated as a parameter of the residual covariance structure conditional on the designated response variable. Consequently, $\lambda$ estimates and partial correlations may vary across alternative model specifications, because the response variable differs among equations(see discussion in ref. 44). Model fit was evaluated using the d-separation method based on Fisher's C-statistic, and model support was compared using the C-statistic Information Criterion corrected for small sample size (CICc). CICc is specifically derived from Fisher's C-statistic and is used in the d-separation framework implemented in phylopath package. Despite differences in how phylogenetic covariance is parameterised, both AICc and CICc share the same interpretative threshold: models with ΔAICc or ΔCICc <2 are considered to have comparable support[78].

## Reporting summary

Further information on research design is available in the Nature Portfolio Reporting Summary linked to this article.

## Data availability

The data used in this study are available in Figshare (https://doi.org/10.6084/m9.figshare.31490812)[79]. The Figshare repository includes the dataset used in the analyses and the phylogenetic trees used for phylogenetic mixed models and for generating Fig. 2. Data underlying Figs. 3 and 4 are provided in Table 1 and Supplementary Table 5. Source Data are provided for Fig. 5 and Supplementary Fig. 4. Source data are provided with this paper.

## Code availability

The code used for data preparation, statistical analyses, and figure generation is available in Figshare (https://doi.org/10.6084/m9.figshare.31490812)[79].

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

## Acknowledgements

We thank Bálint Kovács for extracting some of the data used in the analyses. This work was supported by the National Research, Development and Innovation Office of Hungary to TS (HUN-REN–Debrecen University Reproductive Strategies Research Group, Ref. 1102207, ADVANCED 150852, HU-RIZONT-2024-00109). RPF and TS were supported by the Hungarian Academy of Sciences Guest Professorship scheme 2024-47. AL received funding from the HUN-REN TKI Hungarian Research Network (HUN-REN–PE Evolutionary Ecology Research Group, Ref. 16007) and was also supported by the National Research, Development and Innovation Office of Hungary (ADVANCED 150703).

## Author contributions

T.S. and Z.S. developed the concept of the study and wrote the first draft, Z.S. and A.L. prepared the data, Z.S. analysed the data, supported by A.L. and R.P.F., Y.L. provided feedback on the study design and manuscript, and all authors contributed to revising the manuscript.

## Competing interests

The authors declare no competing interests.
