## [Transparent Peer Review file · Nature Communications]

Demographic causes and social consequences of adult sex ratio variation

Corresponding Author: Dr Zitan Song

Version 0:

Reviewer comments:

Reviewer #1

(Remarks to the Author)

I really enjoyed reading this paper that uses confirmatory path analysis to show that sex differences in mortality and maturation drive adult sex ratios which in turn drive sex differences in mating and parenting rather than the other way around as has also been argued. I'm not familiar with confirmatory path analyses so I cannot judge the statistical side of this paper. I am fairly familiar with the sexual selection literature and so I feel confident in stating that this paper is an important contribution to that literature by testing support for two alternative hypotheses about the evolution of adult sex ratios and breeding systems.

As a demographer, my only concern with the paper is the use of adult mortality rates as if these are constant over adult lifetime which is rarely the case. By assuming a constant adult mortality rate, different estimates will be obtained depending on the age distribution of individuals that happened to be followed/measured and so I wonder how reliable this rate is when used for interspecific comparisons. However, I also appreciate that the authors are limited by what is in the literature. A more robust measure might be something like sex-specific adult lifespan if that is available in the literature? In an ideal world, one would use both a measure of the shape and the pace of life to compare but that would require full age-specific trajectories for all these species. Are any of these birds in the demographic database COMADRE? If there is no way to get age trajectories then we just have to hope there is no consistent bias in the data used but a sentence or two discussing this weakness in the discussion would be good.

Reviewer #2

(Remarks to the Author)

This manuscript is highly relevant, as the authors analyze the interaction between demographic components, sexual selection and adult sex ratio (ASR) based on a comprehensive study of 261 bird species. They also highlight ASR as a crucial component in the evolution of reproductive strategies.

The originality of this work lies in the fact that the authors analyze the relationship between ASR and precopulatory and postcopulatory sexual selection. Furthermore, through confirmatory phylogenetic path analyses, they demonstrate that sexual differences in mortality and maturation drive ASR and that this, in turn, leads to differences in the reproductive mating and breeding system.

Overall, I find this to be a very interesting manuscript, in which the authors have been clear and forceful throughout the manuscript. However, I believe the authors must incorporate or mention in the discussion other relevant factors in ASR variation. Key demographic factors, including migration or permanent residence of the species, and the impact of differential natal dispersal between the sexes, are likely to significantly influence ASR.

Please find below some minor suggestions or queries:

Line 98: Please change tests mass by testes mass.

Results

Line 107: Please improve the resolution of Figure 2; it is currently impossible to interpret.

Discussion

Line 190

I would like to suggest that you must have greater care in the affirmation that the birth sex ratio (BSR) shows no impact on the adult sex ratio. While the results indicate low variability in the BSR, it is important to bear in mind that a small number of samples were analyzed and obtained a low reliability of imputation. Therefore, I consider that the results do not allow conclusively determine the absence of an impact.

Methods

Birth sex ratio

Line 316

Could you explain in more detail what data you considered to obtain the birth sex ratio? How did you combine the hatching and fledging data? How did you account for species with obligate or facultative brood reduction? Please clarify this point further.

Line 353. Mating system

When assigning the scale value, did you consider extra-pair copulations to define the mating system?

Line 371: Please replace Liker et al. 2013 by 58.

Line 454: Please replace Shipley (2016) by 47.

Reviewer #3

(Remarks to the Author)

The paper is very interesting and addresses important questions in evolutionary biology. However, there are some concerns, especially on the methodology as specified in the comments below, which the authors should address to make the manuscript publishable in Nature Communications. I hope my comments below will help the authors to improve this interesting paper.

While the paper is generally easy to read, there are numerous typos and grammatical errors throughout the text. The manuscript would benefit greatly from a thorough revision to address these issues.

Introduction

* The introduction is clearly written and introduces the addressed questions well.

* The authors define Adult Sex Ratio (ASR) as the proportion of males in the population. It would be more precise to specify this as the proportion of males in the sexually mature population.

* Line 64: Fig 1 does not have a and b parts and does not seem related to the text here. Possible missing figure?

* Lines 71-72: The text mentions Hypothesis 1 which is shown in Fig 1, but the figure itself is not split into 'a' and 'b' parts as the text might imply.

* Line 76: Same issue. The text refers to Hypothesis 2 in "fig1b," but there is only a single Figure 1 with no 'a' or 'b' panels.

* Line 88: Potential typo. "the cost of reproductions" should likely be singular: "the cost of reproduction".

* Lines 89 and 99: The text mentions "phylogenetic comparative methods" and "phylogenetic confirmatory path analysis" but provides no references for these methods.

* Line 100: The citation to Fig 1 here seems redundant, as it has already been cited repeatedly in the introduction.

Methods

* Line 304: The text reads "Repeatability = 0.629... by package 'rptR'." It would be better to introduce the software environment, for example: "...as estimated by the R package 'rptR'." The authors have not yet mentioned that R was used for the analyses.

* Line 315: The authors state they combined data for fledgling and hatching sex ratios. For clarity, they should still specify the number of data points from each category (hatching vs. fledgling) in the final dataset.

* Line 327: The authors should specify for how many species they were able to gather data on sex differences in juvenile and adult mortality rates. If this information is in the supplementary materials, it would be good to mention that in the main text as well.

* Pre-copulatory bias data: Similarly, the authors should specify the number of species for which pre-copulatory bias data were gathered. They mention the supplementary materials for sexual size dimorphism data, so they should be consistent and do the same here.

* Line 351: The authors state they used principal components analysis with varimax rotation. They should specify which R package or other software was used to perform this analysis.

* Line 363: The sentence starting "We then take controlled the morphology-to-body size allometries..." is grammatically unclear. The word "take" seems extraneous, "by calculate" should likely be "by calculating", and the phrase "with log-10 transferred" should probably be "after log-10 transformation". The sentence needs to be rewritten for clarity.

* Line 371: The in-text citation "see Liker et al. 2013..." is not consistent with the sequential numeral format used for references throughout the rest of the manuscript, which is the standard for Nature Communications.

* Line 381: Typo. "Among species using in the imputation..." should be "Among species used in the imputation..."

* Line 382: Grammatical error. The sentence starts "o increasing the robustness...", which is likely a typo for "To increase the robustness..."

* Line 389: The meaning of the phrase "deleted as the percentage of the missing data" is unclear. Please rephrase for clarity.

* Line 398: The authors state they fit a "phylogenetically controlled mixed model." They should mention and cite the specific R package(s) used to run this analysis.

* Line 401: The phrase "Since the limitation effect of birth sex ratio on ASR..." is grammatically awkward. Did the authors mean "Since the limited effect...?"

- * Line 402: The text "...from follow analysis" should be corrected to "...from following analyses."
- * Line 403: The authors state they "averaged the standardized values of juvenile mortality bias, maturation bias, and adult mortality bias" to create a composite "demography bias" variable. The justification for this approach is unclear. It is not immediately obvious how averaging these disparate values provides a meaningful measure of the overall demography bias effect. The authors should provide a rationale for this methodological choice.
- * Line 408: A similar issue arises here. The authors state they "considered an aggregate of standardized variables" for pre-copulatory bias (sexual size dimorphism, plumage dimorphism, mating system). The text should clarify how these variables were aggregated and provide a clear justification for this specific methodological approach.
- * Line 409: The sentence describing the PCA for plumage dimorphism is awkwardly phrased. I suggest changing it to something like: "For plumage dimorphism, we applied a Principal Component Analysis (PCA) to reduce the dimensionality of the plumage dimorphism values from five body regions and extracted the first Principal Component (PC1)."
- * Line 413: The authors state, "for post-copulatory bias, we focused on relative testes mass and extra-pair paternity." They should briefly explain the rationale behind choosing these two specific variables to represent post-copulatory bias.
- * Line 413: The statement "All these variables were averaged to represent their respective mate choice categories" is unclear. It is not specified which "variables" are being averaged, and the justification for averaging variables that may be on different scales or represent fundamentally different biological aspects is missing. This requires a more detailed explanation and rationale.
- * Line 414: "we use first PC of female participated value of brooding and chick attending." This sentence is awkwardly phrased. A clearer wording would be: "For parental care bias, we used the first Principal Component (PC1) from a PCA on the values for female participation in brooding and chick attending."
- * Line 426: The title "Path analyses" should be changed to "Phylogenetic Path Analyses"
- * Lines 446-455 (Phylogenetic Path Analyses Section): The authors correctly cite von Hardenberg and Gonzalez-Voyer (2013), but then proceed to use an older method based on Phylogenetic Independent Contrasts (PICs). Why did the authors not use the PPA method from von Hardenberg and Gonzalez-Voyer (2013), as implemented in the phyloPath package (Van der Bijl, 2018), which uses PGLS and accounts for phylogenetic non-independence in a single, integrated step? The authors should provide a clear justification for their choice of the PIC-based method.
- * Line 457: The use of standard Structural Equation Modeling (SEM) fit indices (CFI, TLI, RMSEA, SRMR) via the lavaan package to assess model fit is a significant concern. These indices and the lavaan package are designed for models with independent data points. However, the data used here are phylogenetic and thus violate the core assumption of independence. While the authors have attempted to control for phylogeny using PICs, the appropriateness of using standard SEM fit statistics to PICs has not been rigorously tested to my knowledge. The underlying assumptions of the fit indices are possibly violated. The authors must provide a strong rationale for why this approach is valid in a phylogenetic context, or else reconsider their method for assessing model fit.
- * Lines 461-465 The authors mention replicating their analyses across 100 randomly selected phylogenetic trees, but they have not stated anywhere in the Methods section what the source of their primary phylogeny is. This is a critical piece of methodological information that must be included.

Results

- * Line 144: A variance inflation factor (VIF) analysis is mentioned in the Results but not described in the Methods.
- * Line 154: The authors refer to "AICc" here, but they refer to "CICc" in the Methods section (line 458). They should use the correct and consistent terminology throughout the manuscript. Given the context of PPA, CICc is the appropriate criterion.

Discussion

- * Lines 176-177: The phrasing "Using the most comprehensive analyses..." comes across as an overstatement. A more objective tone is recommended.
- * Lines 205-210: The authors note that the correlation between parental care and ASR disappears when other variables are included. This statistical result strongly suggests collinearity from common underlying causes. This is precisely the kind of complex relationship that a well-specified PPA is designed to disentangle, which would be a more powerful approach than simply noting the changing partial correlation.
- * Line 253: The authors state that "path analyses may fail to detect" evolutionary feedbacks. They should be more precise about why this is the case. The limitation arises because Phylogenetic Path Analysis, is based on Directed Acyclic Graphs (DAGs). By definition, DAGs cannot include cycles, which are required to model reciprocal relationships or feedback loops. Therefore, while PPA is excellent at comparing alternative causal scenarios, it is structurally unsuited for testing hypotheses that involve direct evolutionary feedback.
- * Line 288: The final sentence makes a compelling point about Darwin's ideas but cites an online popular science article. It might be beneficial to better explain what his idea about the link between sexual selection and sex ratios was and also cite the primary source for Darwin's original hypothesis, and perhaps the peer-reviewed research that the popular article discusses, to better ground this concluding thought in the primary literature.

van der Bijl, W. (2018). phyloPath: an R package for phylogenetic path analysis. PeerJ, 6, e4718.

Version 1:

Reviewer comments:

Reviewer #1

(Remarks to the Author)

The authors have adequately addressed the issue I raised considering the limitations of available data. I'm happy with the manuscript as it is now.

Reviewer #2

(Remarks to the Author)

In this new version, the authors have demonstrated great effort and dedication. I believe they have responded appropriately and carefully to each of my observations and comments, which is reflected in a much clearer and more precise article.

Reviewer #3

(Remarks to the Author)

While I appreciate the authors' detailed explanations, I remain unconvinced about the theoretical justification for prioritising the "pre-transformation" approach (Santos/Liker method) over the residual based PGLS approach for Phylogenetic Path Analysis (von Hardenberg and Gonzalez-Voyer, 2013).

The authors state that they estimate λ separately for each variable, before computing contrasts. However, as established in the literature for PGLS (e.g., Freckleton et al. 2002; Revell 2010), λ is a parameter of the residual error structure, not the raw traits. Pre-transforming variables independently assumes that the phylogenetic signal in the response is independent of the predictor, an assumption rarely met in co-evolving biological systems.

Furthermore, the authors argue that the von Hardenberg & Gonzalez-Voyer (2013) method is inconsistent because the direction of the regression ($X \rightarrow Y$ vs. $Y \rightarrow X$) yields different λ values. I would argue that this is not a lack of robustness, but a biological requirement of path analysis. Path analysis is a test of directed causal hypotheses; since λ accounts for the unexplained variance in the response, it should change if the hypothesized response variable changes. Forcing symmetry via univariate transformation bypasses this conditional logic leading possibly to inflated type I error rates.

The use of standard SEM fit indices (CFI, TLI, RMSEA) on λ -transformed PICs remains, in my view, problematic. These indices were developed for independent data sampled from a population. As noted in Freckleton (2009), phylogenetic data, even when transformed, often do not possess the same number of effective degrees of freedom as independent data points ($N-1$). Standard SEM software is 'blind' to the hierarchical source of the contrasts and may therefore produce over-confident fit statistics based on an inflated denominator of independence. To my knowledge, the performance of Global Fit Indices (like CFI and TLI) has not been rigorously validated or even extensively tested on phylogenetically structured contrasts (see Gonzalez-Voyer & von Hardenberg 2014). The 'null' model used to calculate CFI/TLI in lavaan assumes a specific covariance structure that may not be biologically or statistically appropriate when the data origin is a bifurcating tree, even after a λ transformation. Furthermore, even if the contrasts are independent, they may not be identically distributed if there is heteroscedasticity introduced by the tree topology or if the univariate λ values do not perfectly capture the co-evolutionary signal in the multivariate system.

While it is encouraging that the results are concordant across methods, I strongly suggest the authors move the results obtained with phylopath (which implements the von Hardenberg and Gonzalez-Voyer, 2013 method) to the main text as the primary analysis rather than in the Suppl. Materials. If the authors decide to still keep also the Santos/Liker analysis, I would recommend the inclusion of an acknowledgement that the pre-transformation method, while providing similar results as the PPA approach, is a pragmatic alternative which may however not fully account for co-evolutionary signals in the residuals.

On a minor point, the authors state that they used a "more up-to-date path analysis implementation (piecewiseSEM)". I would like a clarification on this: PiecewiseSEM implements the exact same d-sep approach as in phylopath, with the only difference that it does not include PGLS models.

Minor comments in the main text:

Line 498 and subsequent: the common definition is "Phylogenetic Path Analysis", and not "Analyses"

Lines 547-49. You can cite Shipley (2013), which demonstrates the equivalence of AIC (based on the likelihood) with CIC (based on the C statistic).

References

Freckleton, R. P. (2009). The seven deadly sins of comparative analysis. *Journal of Evolutionary Biology*, 22(3), 459-467.

Freckleton, R. P., Harvey, P. H., & Pagel, M. (2002). Phylogenetic analysis and comparative data: a test and review of evidence. *The American Naturalist*, 160(6), 712-726.

Gonzalez-Voyer, A., & von Hardenberg, A. (2014). An Introduction to Phylogenetic Path Analysis. In: *Modern Phylogenetic Comparative Methods and Their Application in Evolutionary Biology* (pp. 201-229). Springer, Berlin, Heidelberg.

Revell, L. J. (2010). Phylogenetic signal and linear regression on species data. *Methods in Ecology and Evolution*, 1(4),

319-329.

Shiple, B. (2013). The AIC model selection method applied to path analytic models compared using a d-separation test. *Ecology*, 94(3), 560-564.

REVIEWER COMMENTS

Reviewer #1 (Remarks to the Author):

I really enjoyed reading this paper that uses confirmatory path analysis to show that sex differences in mortality and maturation drive adult sex ratios which in turn drive sex differences in mating and parenting rather than the other way around as has also been argued. I'm not familiar with confirmatory path analyses so I cannot judge the statistical side of this paper. I am fairly familiar with the sexual selection literature and so I feel confident in stating that this paper is an important contribution to that literature by testing support for two alternative hypotheses about the evolution of adult sex ratios and breeding systems.

Many thanks for this positive overall assessment. As you will see below and in the manuscript, your comments led us to rethink our approach and check various alternatives.

As a demographer, my only concern with the paper is the use of adult mortality rates as if these are constant over adult lifetime which is rarely the case. By assuming a constant adult mortality rate, different estimates will be obtained depending on the age distribution of individuals that happened to be followed/measured and so I wonder how reliable this rate is when used for interspecific comparisons. However, I also appreciate that the authors are limited by what is in the literature. A more robust measure might be something like sex-specific adult lifespan if that is available in the literature? In an ideal world, one would use both a measure of the shape and the pace of life to compare but that would require full age-specific trajectories for all these species. Are any of these birds in the demographic database COMADRE? If there is no way to get age trajectories then we just have to hope there is no consistent bias in the data used but a sentence or two discussing this weakness in the discussion would be good.

We appreciate this thoughtful and constructive comment. We fully agree that adult mortality could be age-dependent, and that sex-biased senescence may occur. Within the limits of the available literature, we have addressed this concern in three key ways and clarified the changes in the manuscript.

(1) Data extraction from the same population to minimise differential age-structure bias.

Male and female adult mortality estimates in this work were taken from the same population and study. This reduces the risk that between-study differences in age composition inflate sex contrasts and thus limits the influence of sex-biased senescence. In addition, prime breeders in most populations are relatively young, so the age classes that contribute most to ASR are those least affected by senescence. This further limits the potential impact of sex-biased senescence on the sex-biased mortality estimates we compiled.

(2) Methodological heterogeneity and sensitivity analyses.

Our dataset includes population estimates derived using five approaches: (i) capture-mark-recapture (CMR), (ii) return rate, (iii) ringing-recovery analysis, (iv) radio-telemetry (known-fate) studies, and (v) population age-distribution analysis. Because some approaches are particularly susceptible to under-estimating for age (particularly return-rate and age-distribution methods, which lack individual age tracking or rely on strong stable-age assumptions), we conducted robustness checks in two steps:

First, we conservatively grouped CMR as “high-quality” and pooled all non-CMR estimates (ii–v) as “other” to test whether method group affects mortality estimates in species with multiple populations. We analysed 73 population-level records across 35 species (≥ 2 populations per species). Method group had no detectable effect on estimated adult mortality (see Table R1).

Table R1. Effect of mortality estimation method on adult mortality rates in species with multiple populations. We analysed 73 population-level estimates across 35 species, classifying capture-mark-recapture (CMR) as “high-quality” (n = 42) and pooling other approaches (return rate, ringing-recovery, radio-telemetry, age-distribution) as “low-quality/other”. Posterior summaries are shown for models of male and female adult mortality (intercept = CMR).

	Posterior mean	Lower – upper 95% CI	pMCMC
Male mortality rate			
Intercept	0.290	0.118 – 0.452	0.002
Measurement (low)	0.029	-0.023 – 0.092	0.326
Female mortality rate			
Intercept	0.305	0.128 – 0.474	<0.001
Measurement (low)	0.049	-0.010 – 0.107	0.112

Second, for species with 2–5 independently sampled populations (n = 35 species), adult mortality showed high repeatability across populations (male: 0.733; female: 0.678). This cross-population consistency indicates that, within species, sex differences in adult mortality are not strongly influenced by population-level variation in age structure or sex-biased senescence. We have added this results in Lines 381-383.

(3) Sex-specific adult lifespan.

We agree that sex-specific adult lifespan would be a valuable complementary measure, but such data are very limited for birds. We checked COMADRE and found that among 3,550 animal population matrices (445 bird species), only four bird species included both sexes, with no overlap with our dataset. Beyond COMADRE, we identified three additional sources reporting sex-biased lifespan (Clutton-Brock & Isvaran 2007, Che-Castaldo et al., 2019, Carey & Judge 2000)^{1–3} and found 16 species overlapping with our dataset. In this subset, sex-biased adult lifespan, which calculated as lifespan minus mature age, explained limited variation in ASR relative to sex-biased adult mortality (Table S10). We have added this analysis to the Supplementary Material and discuss it in the manuscript (Lines 205–216).

We also explored this idea with the most up-to-date sex-specific lifespan data compiled from captive populations (Staerk et al., 2025)⁴. Across 55 overlapping species, the difference in adult life expectancy between females and males showed no significant association with ASR (PGLS: estimate \pm SD = -0.115 ± 0.114 , $t = -1.013$, $p = 0.316$).

Because these values derive from captive populations, we did not retain this analysis in the main text or Supplementary Material. Overall, these checks are consistent but not definitive with our inference that sex differences in adult survival, rather than longevity *per se*, dominate cross-species variation in ASR.

We suggest the following reasons why sex-biased lifespan has limited explanatory power for ASR, whereas sex-biased adult mortality does:

- a) Temporal sensitivity mismatch: Under a stable-age framework, ASR primarily reflects sex differences in survival at young and prime breeding ages, which dominate the adult census. In contrast, adult lifespan averages hazards over the entire adult span and is disproportionately influenced by the long-lived tail. Consequently, lifespan sex differences have weaker leverage on ASR than sex-specific adult mortality.**
- b) Metric heterogeneity: Published “lifespan” varies in definition and method (e.g., maximum vs. mean/median life expectancy; wild vs. mixed/captive sources), adding noise that dilutes any ASR signal.**
- c) Estimation constraints: Robust lifespan requires following full adult cohorts to death (with heavy censoring, emigration, and detection issues), whereas annual adult mortality is routinely estimable from longitudinal mark–recapture and is directly comparable across studies.**

Following the Reviewer’s suggestion, we added a brief discussion of sex-biased adult lifespan in the main text (Lines 205-216) and added the exploratory analyses to the Supplementary Material (Table S10). Given limited and heterogeneous coverage, these analyses add little beyond adult mortality, so we retained the latter as the primary predictor.

Reviewer #2 (Remarks to the Author):

This manuscript is highly relevant, as the authors analyze the interaction between demographic components, sexual selection and adult sex ratio (ASR) based on a comprehensive study of 261 bird species. They also highlight ASR as a crucial component in the evolution of reproductive strategies.

The originality of this work lies in the fact that the authors analyze the relationship between ASR and precopulatory and postcopulatory sexual selection. Furthermore, through confirmatory phylogenetic path analyses, they demonstrate that sexual differences in mortality and maturation drive ASR and that this, in turn, leads to differences in the reproductive mating and breeding system.

We appreciate this evaluation and have considered each point in our revision (see below).

Overall, I find this to be a very interesting manuscript, in which the authors have been clear and forceful throughout the manuscript. However, I believe the authors must incorporate or mention in the discussion other relevant factors in ASR variation. Key demographic factors, including migration or permanent residence of the species, and the impact of differential natal dispersal between the sexes, are likely to significantly influence ASR.

Thank you for this insightful suggestion. We fully agree that additional processes may modulate ASR. In the revision, we (i) expanded the Discussion to address migration and sex-biased natal dispersal (Discussion, lines 217–232), and (ii) added two supplementary tests to check these factors. Both analyses showed no detectable effects on ASR or on population demographic bias (Tables S11 & S12).

Migration:

We coded all species as resident, partially migratory, or fully migratory. ASR and demographic biases did not differ among migration categories (Table S11).

Sex-biased dispersal:

Using the two most up-to-date compilations of sex-specific natal dispersal distances (Fandos et al. 2023, Végvári et al. 2018)^{5,6}, we assembled 63 species with male and female distances. Dispersal bias (log male – log female) showed no effect on ASR and demographic biases (Table S12), indicating no impact on our main conclusions. Data limitations precluded separating natal from breeding dispersal broadly: Fandos et al. 2023 report sex-averaged distances without separating natal vs. breeding for male and female, and Végvári et al. 2018 include only 14 species for breeding dispersal. Following prior work, we therefore analysed the available average distances (Supplementary Table S12).

Because of these data limitations and given that both analyses consistently showed no detectable effects on ASR or population demographic bias, we have retained the corresponding results in the Supplementary Material and summarise their implications in the main text (Discussion, lines 217–232).

Please find below some minor suggestions or queries:

Line 98: Please change tests mass by testes mass.

Thank you, corrected (line 102).

Results

Line 107: Please improve the resolution of Figure 2; it is currently impossible to interpret.

Thank you for the helpful suggestion. We re-generated Figure 2 to address both resolution and readability, including changing the phylogenetic layout from a circular to a rectangular tree to improve clarity.

Discussion

Line 190

I would like to suggest that you must have greater care in the affirmation that the birth sex ratio (BSR) shows no impact on the adult sex ratio. While the results indicate low variability in the BSR, it is important to bear in mind that a small number of samples were analyzed and obtained a low reliability of imputation. Therefore, I consider that the results do not allow conclusively determine the absence of an impact.

We appreciate this point and revised the discussion to adopt a more cautious tone regarding the interpretation of BSR's impact on ASR in Lines 198-200. Specifically, we now clarify that the absence of a detectable effect may reflect both the limited variance in BSR and the relatively small number of species with reliable BSR data.

Methods

Birth sex ratio

Line 316

Could you explain in more detail what data you considered to obtain the birth sex ratio? How did you combine the hatching and fledging data? How did you account for species with obligate or facultative brood reduction? Please clarify this point further.

Thank you: The revised MS clarifies the data source and handling of birth sex ratio (BSR) data in the Methods section (Lines 361–368). Specifically, BSR was primarily based on fledgling sex ratio data (n = 63), supplemented with hatchling sex ratio data (n = 40). We prioritized fledgling data because it is more closely related to ASR outcomes, and information on sex-biased nestling mortality is generally lacking across species.

Regarding potential biases due to brood reduction, most of the hatchling data in our dataset come from avian orders such as Anseriformes (n = 9), Charadriiformes (n = 10), and Passeriformes (n = 13), in which obligate or facultative brood reduction is rare or absent. Therefore, we believe that brood reduction effects on our BSR estimates are limited.

Line 353. Mating system

When assigning the scale value, did you consider extra-pair copulations to define the mating system?

We appreciate this point. Note that we used data on social mating system in the MS since these data were lot more widely available than extra-pair copulations (EPC) or data on extra-pair paternity (EPP). This choice aligns with our framework's focus on parental investment and reproductive behaviour. We have clarified this in the Methods (lines 413–417), noting that our usage is consistent with prior work and our own previous studies (e.g., Gonzalez-Voyer et al., 2022; Liker & Székely, 2005).

Line 371: Please replace Liker et al. 2013 by 58.

Thank you: corrected (Line 427).

Line 454: Please replace Shipley (2016) by 47.

Thank you: corrected (Line 527).

Reviewer #3 (Remarks to the Author):

The paper is very interesting and addresses important questions in evolutionary biology. However, there are some concerns, especially on the methodology as specified in the comments below, which the authors should address to make the manuscript publishable in Nature Communications. I hope my comments below will help the authors to improve this interesting paper.

We appreciate this positive overall assessment and have considered each point in our revision (see below).

While the paper is generally easy to read, there are numerous typos and grammatical errors throughout the text. The manuscript would benefit greatly from a thorough revision to address these issues.

Thank you. We weeded out numerous typos and errors in the revised the manuscript.

Introduction

* The introduction is clearly written and introduces the addressed questions well.

Thank you for your positive assessment.

* The authors define Adult Sex Ratio (ASR) as the proportion of males in the population. It would be more precise to specify this as the proportion of males in the sexually mature population.

Thank you for the helpful suggestion. We define ASR as the proportion of males among adults and usually avoid the phrase “sexually mature population” because it can be confused with (i) the sex ratio at maturation (MSR)—the ratio at the moment individuals enter adulthood—and (ii) the operational sex ratio (OSR)—the ratio of currently receptive breeders, which all have strong influence on sex role⁷. Our wording clarifies that ASR refers to all adults that have attained maturity, regardless of whether they are presently breeding. Therefore, we prefer to retain the established definition used in the ASR literature. We hope this addresses your concern, but we are happy to adjust if you think further clarification would help.

* Line 64: Fig 1 does not have a and b parts and does not seem related to the text here. Possible missing figure?

Thank you for pointing this out. We have now added panel labels (a) and (b) to Fig. 1 for clarity. Since the content related to Fig. 1a is already present at the end of the first paragraph, we have removed the earlier reference at Line 64 to avoid redundancy. All in-text figure references have been updated to align with the revised figure layout.

* Lines 71-72: The text mentions Hypothesis 1 which is shown in Fig 1, but the figure itself is not split into 'a' and 'b' parts as the text might imply.

We appreciate you pointing out this discrepancy. We have added panel labels (a) and (b) to Fig. 1, which now clearly distinguishes the visual representations of Hypotheses 1 and 2.

* Line 76: Same issue. The text refers to Hypothesis 2 in "fig1b," but there is only a single Figure 1 with no 'a' or 'b' panels.

We appreciate your careful reading. The panel label (b) has now been added to Fig. 1 to correspond with the textual reference to Hypothesis 2.

* Line 88: Potential typo. "the cost of reproductions" should likely be singular: "the cost of reproduction".

Thank you: corrected, see line 92.

* Lines 89 and 99: The text mentions "phylogenetic comparative methods" and "phylogenetic confirmatory path analysis" but provides no references for these methods.

Thank you; references added see lines 93 and 103, respectively.

* Line 100: The citation to Fig 1 here seems redundant, as it has already been cited repeatedly in the introduction.

Thank you: reference was removed.

Methods

* Line 304: The text reads "Repeatability = 0.629... by package 'rptR'." It would be better to introduce the software environment, for example: "...as estimated by the R package 'rptR'." The authors have not yet mentioned that R was used for the analyses.

Thank you for the suggestion. To maintain a logical flow, we begin the Methods section with data collection and introduce the R environment at the start of the Statistical Analyses subsection. However, we agree that R should be mentioned earlier, particularly when describing the repeatability estimates and data imputation. We have now clarified that these steps were conducted in R, and explicitly noted the software environment at Lines 351, 383, 406, 432 and 443.

* Line 315: The authors state they combined data for fledgling and hatching sex ratios. For clarity, they should still specify the number of data points from each category (hatching vs. fledgling) in the final dataset.

Thank you for the suggestion. In the submitted dataset, we now include separate columns for hatching sex ratio and fledging sex ratio, along with their corresponding references. In our analyses, we used fledging sex ratios whenever available, and supplemented them with hatching sex ratios when fledging data were not available. We have also added a column called "BSR_note" in the dataset to indicate the source of each birth sex ratio (BSR) value. This clarification has been added to Lines 361–368 in the revised manuscript.

* Line 327: The authors should specify for how many species they were able to gather data on sex differences in juvenile and adult mortality rates. If this information

is in the supplementary materials, it would be good to mention that in the main text as well.

Thank you for the suggestion. We have added this information in Line 377-383.

* Pre-copulatory bias data: Similarly, the authors should specify the number of species for which pre-copulatory bias data were gathered. They mention the supplementary materials for sexual size dimorphism data, so they should be consistent and do the same here.

Thank you for the suggestion. We have now added the sample size for all pre and post-copulatory bias data in the method at line: 395, 399, 413, 420 and 425.

* Line 351: The authors state they used principal components analysis with varimax rotation. They should specify which R package or other software was used to perform this analysis.

Thank you for pointing this out. We performed the principal component analysis with varimax rotation using the *principal()* function from the psych R package. We have now added this information to Line 406 of the Methods section.

* Line 363: The sentence starting "We then take controlled the morphology-to-body size allometries..." is grammatically unclear. The word "take" seems extraneous, "by calculate" should likely be "by calculating", and the phrase "with log-10 transferred" should probably be "after log-10 transformation". The sentence needs to be rewritten for clarity.

Thank you for pointing this out. We have revised the sentence for clarity in Line 420. It now reads: "To control for morphology-to-body size allometries, we calculated residual testes mass by regressing log₁₀-transformed testes mass against log₁₀ transferred male body mass."

* Line 371: The in-text citation "see Liker et al. 2013..." is not consistent with the sequential numeral format used for references throughout the rest of the manuscript, which is the standard for Nature Communications.

Thank you for pointing this out. We have corrected the citation in Line 427.

* Line 381: Typo. "Among species using in the imputation..." should be "Among species used in the imputation..."

Thank you for pointing this out. We have corrected this in Line 436.

* Line 382: Grammatical error. The sentence starts "o increasing the robustness...", which is likely a typo for "To increase the robustness..."

Thank you for pointing this out. We have corrected this in Line 437.

* Line 389: The meaning of the phrase "deleted as the percentage of the missing data" is unclear. Please rephrase for clarity.

Thank you for the comment. We have clarified the description of the imputation procedure in Lines 443–447, as the previous wording may have been misleading. Specifically, we used a systematic leave-one-out cross-validation approach, in which each original (non-missing) data point was removed once, re-imputed using the same model, and compared to its true value. We have also updated the legend of Table S3 to reflect this clarification.

* Line 398: The authors state they fit a "phylogenetically controlled mixed model." They should mention and cite the specific R package(s) used to run this analysis.

Thank you for pointing this out. We have added the reference in Line 454.

* Line 401: The phrase "Since the limitation effect of birth sex ratio on ASR..." is grammatically awkward. Did the authors mean "Since the limited effect...?"

Thank you for pointing this out. We have revised the sentence for clarity in Line 457.

* Line 402: The text "...from follow analysis" should be corrected to "...from following analyses."

Thank you for pointing this out. We have revised the sentence for clarity in Line 458.

* Line 403: The authors state they "averaged the standardized values of juvenile mortality bias, maturation bias, and adult mortality bias" to create a composite "demography bias" variable. The justification for this approach is unclear. It is not immediately obvious how averaging these disparate values provides a meaningful measure of the overall demography bias effect. The authors should provide a rationale for this methodological choice.

Thank you for the comment. We have added an explanation in Lines 458–464 of the revised manuscript. By standardizing we adjusted the variables to the same scale and by averaging the three life-stage-specific bias measures we aimed to capture an integrative measure of the demographic asymmetry between the sexes influencing ASR.

* Line 408: A similar issue arises here. The authors state they "considered an aggregate of standardized variables" for pre-copulatory bias (sexual size dimorphism, plumage dimorphism, mating system). The text should clarify how these variables were aggregated and provide a clear justification for this specific methodological approach.

Thank you for the comment. We have revised and clarified the relevant sentence in Lines 469–470 and added an explanation for the averaging approach used for both pre- and post-copulatory bias indices in Lines 478–482.

* Line 409: The sentence describing the PCA for plumage dimorphism is awkwardly phrased. I suggest changing it to something like: "For plumage dimorphism, we applied a Principal Component Analysis (PCA) to reduce the dimensionality of the

plumage dimorphism values from five body regions and extracted the first Principal Component (PC1).”

Thank you for the suggestion. We have revised the sentence in Line 471-473 following your recommendation.

* Line 413: The authors state, "for post-copulatory bias, we focused on relative testes mass and extra-pair paternity." They should briefly explain the rationale behind choosing these two specific variables to represent post-copulatory bias.

Thank you for the thoughtful comment. We have clarified this in the revised manuscript in Line 473-478. While testes mass reflects male physiological investment in sperm competition, EPP captures realized reproductive outcomes of post-copulatory sexual selection which also including female choice. Averaging the standardized values these two measures provides a sex-inclusive and integrative index of post-copulatory bias.

* Line 413: The statement "All these variables were averaged to represent their respective mate choice categories" is unclear. It is not specified which "variables" are being averaged, and the justification for averaging variables that may be on different scales or represent fundamentally different biological aspects is missing. This requires a more detailed explanation and rationale.

Thank you for point this out. We were referring to the fact that pre-copulatory bias was calculated as the mean of standardized values for sexual size dimorphism, sexual plumage dimorphism (PC1), and mating system, and post-copulatory bias was calculated as the mean of standardized values for relative testes mass and extra-pair paternity. Together with standardized parental-care bias (PC1) and demographic bias (the mean of standardized values for juvenile mortality bias, maturation bias, and adult mortality bias), these indices allow direct comparison of effect sizes among components in models relating bias indices to ASR. We have added this explanation in the revised Methods (lines 479–483).

* Line 414: "we use first PC of female participated value of brooding and chick attending." This sentence is awkwardly phrased. A clearer wording would be: "For parental care bias, we used the first Principal Component (PC1) from a PCA on the values for female participation in brooding and chick attending."

Thank you for the suggestion. We have revised the sentence in Lines: 483-485.

* Line 426: The title "Path analyses" should be changed to "Phylogenetic Path Analyses"

Thank you! Changed as suggested in Line 498.

* Lines 446-455 (Phylogenetic Path Analyses Section): The authors correctly cite von Hardenberg and Gonzalez-Voyer (2013), but then proceed to use an older method based on Phylogenetic Independent Contrasts (PICs). Why did the authors not use the PPA method from von Hardenberg and Gonzalez-Voyer (2013), as implemented in the phyloPath package (Van der Bijl, 2018), which uses PGLS and

accounts for phylogenetic non-independence in a single, integrated step? The authors should provide a clear justification for their choice of the PIC-based method.

Thank you for pointing this out. We apologize for the mistaken citation in the original version. We did not use the von Hardenberg and González-Voyer (2013) method in the original version but cited it erroneously; we have now removed the citation at line 499.

In our original version of this manuscript, an approach that extends Santos' (2012) method (see Liker et al 2021) was used for PPA because this approach satisfies the assumptions of path analysis better than an alternative method based on phylogenetic regressions proposed by von Hardenberg and Gonzalez-Voyer (2013). This latter approach is not robust to changes in the specification of the model: if variable Y is regressed on X and λ estimated, then the estimates of the partial correlations and λ may be different from those obtained when X is regressed on Y with λ estimated (for a detailed discussion see the Appendix of Liker et al. 2021 Evolution). The approach we have taken avoids this inconsistency and also uses a more up-to-date path analysis implementation (piecewise SEM). Please also note that although we used the PIC function of the *ape* R package for phylogenetic transformation, this is not identical using phylogenetic contrasts because this latter method assume complete phylogenetic dependence in the data ($\lambda=1$) whereas we estimated lambda separately for each variable from the data prior to computing the contrasts.

Additionally, to test the robustness of our PPA results, we repeated the analyses using the approach implemented in *phylopath*, and present the results in Line 169-173 and Table S9. The outcomes of this latter PPA were nearly identical to those we obtained from our original analysis, further supporting the validity of our conclusions. We also describe the rationale for applying both PPA approaches and illustrate their differences in the Methods section (Lines 540-549).

* Line 457: The use of standard Structural Equation Modeling (SEM) fit indices (CFI, TLI, RMSEA, SRMR) via the lavaan package to assess model fit is a significant concern. These indices and the lavaan package are designed for models with independent data points. However, the data used here are phylogenetic and thus violate the core assumption of independence. While the authors have attempted to control for phylogeny using PICs, the appropriateness of using standard SEM fit statistics to PICs has not been rigorously tested to my knowledge. The underlying assumptions of the fit indices are possibly violated. The authors must provide a strong rationale for why this approach is valid in a phylogenetic context, or else reconsider their method for assessing model fit.

Thank you for pointing this out. We do not apply lavaan to raw phylogenetic data. Instead, for each trait we first estimate Pagel's λ by maximum likelihood, rescale the tree accordingly, and then compute phylogenetically independent contrasts (PICs) on the λ -rescaled unit tree. As noted above, this is not identical to classical phylogenetic contrasts which assume Pagel's λ fixed at 1 (i.e., the full Brownian-motion covariance given the tree), whereas we estimate λ separately for each variable and transform the tree before computing contrasts. Under the standard Brownian/ λ model, the resulting contrasts are independent and identically distributions (i.i.d – e.g. see Garland & Ives

2000; Freckleton 2012) with constant variance which fully satisfies the assumptions of the analysis.

Importantly, we did not rely on a single framework to assess model fit or draw conclusions. We analysed the same causal structures using three complementary approaches—piecewiseSEM and lavaan applied to λ -transformed PICs, and the phylogenetic path-analysis framework in phylopath—and all three approaches identified the same best models and yielded concordant conclusions. This consistency across methods supports the robustness of our results and, we hope, alleviates the reviewer’s concern.

* Lines 461-465 The authors mention replicating their analyses across 100 randomly selected phylogenetic trees, but they have not stated anywhere in the Methods section what the source of their primary phylogeny is. This is a critical piece of methodological information that must be included.

We thank the reviewer for pointing out this omission. We have now added a detailed description of the source of the phylogenetic trees used in our analysis to the Methods Line 534-536.

Results

* Line 144: A variance inflation factor (VIF) analysis is mentioned in the Results but not described in the Methods.

We thank the reviewer for pointing out this omission. We have now described the VIF analysis to the Methods Line 467-469.

* Line 154: The authors refer to "AICc" here, but they refer to "CICc" in the Methods section (line 458). They should use the correct and consistent terminology throughout the manuscript. Given the context of PPA, CICc is the appropriate criterion.

We thank the reviewer for pointing out this mistake. We have clarified the distinction between AICc and CICc in the main text (Lines 529–530 and 544–549). AICc is appropriate for models based on maximum likelihood estimation, such as our PIC-based approach, while CICc is specifically derived from Fisher’s C-statistic and used in the d-separation framework implemented in ‘*phylopath*’. We have corrected the terminology in Tables S7 and S8, where “AICc” was previously mislabeled due to a typographical error. Since we also applied the phylopath method for robustness, we report CICc values in Table S9.

Discussion

* Lines 176-177: The phrasing "Using the most comprehensive analyses..." comes across as an overstatement. A more objective tone is recommended.

We thank the reviewer for this helpful suggestion. We agree that the original phrasing may have appeared overstated. We have revised this sentence in Line 180.

* Lines 205-210: The authors note that the correlation between parental care and

ASR disappears when other variables are included. This statistical result strongly suggests collinearity from common underlying causes. This is precisely the kind of complex relationship that a well-specified PPA is designed to disentangle, which would be a more powerful approach than simply noting the changing partial correlation.

We thank the reviewer for this insightful comment. As suggested, we now discuss the possibility that the correlation between parental care and ASR is mediated by pre-copulatory mate choice, based on our phylogenetic path analysis results (Lines 240–250). Moreover, the disappearance of the ASR–parental care correlation in multivariate models likely reflects an indirect relationship rather than multicollinearity, as indicated by the combined results of our VIF analysis and phylogenetic path analysis. We believe this strengthens our interpretation.

* Line 253: The authors state that "path analyses may fail to detect" evolutionary feedbacks. They should be more precise about why this is the case. The limitation arises because Phylogenetic Path Analysis, is based on Directed Acyclic Graphs (DAGs). By definition, DAGs cannot include cycles, which are required to model reciprocal relationships or feedback loops. Therefore, while PPA is excellent at comparing alternative causal scenarios, it is structurally unsuited for testing hypotheses that involve direct evolutionary feedback.

We thank the reviewer for this helpful clarification. We have made it more clear about that phylogenetic path analysis is based on directed acyclic graphs, which by definition cannot represent reciprocal causation or feedback loops. We have revised the sentence in Line 295 to state this structural limitation more precisely.

Furthermore, we clarified that our analysis focuses on comparing alternative unidirectional causal pathways among demographic traits, ASR, and mating systems—an appropriate application of PPA. We also note that discrepancies with previous studies may stem from differences in model structure: whereas earlier work focused solely on maturation bias, we employed a more integrative approach that included maturation and mortality as components of population demography, along with both pre- and post-copulatory sexual selection. These points have now been clarified in the revised text (Lines 298–303).

* Line 288: The final sentence makes a compelling point about Darwin's ideas but cites an online popular science article. It might be beneficial to better explain what his idea about the link between sexual selection and sex ratios was and also cite the primary source for Darwin's original hypothesis, and perhaps the peer-reviewed research that the popular article discusses, to better ground this concluding thought in the primary literature.

We appreciate this comment and now we quote Darwin's original text (Line 54-56) and added a citation to his idea in the revised MS.

References

1. Clutton-Brock, T. H. & Isvaran, K. Sex differences in ageing in natural populations of vertebrates. *Proceedings of the Royal Society B: Biological Sciences* **274**, 3097–3104 (2007).
2. Che-Castaldo, J. P., Byrne, A., Perišin, K. & Faust, L. J. Sex-specific median life expectancies from ex situ populations for 330 animal species. *Sci Data* **6**, 190019 (2019).
3. Carey, J. R. . & Judge, D. S. . *Longevity Records: Life Spans of Mammals, Birds, Amphibians, Reptiles, and Fish*. (Odense University Press, Odense, 2000).
4. Staerk, J. *et al.* Sexual selection drives sex difference in adult life expectancy across mammals and birds. *Sci Adv* **11**, (2025).
5. Végvári, Z. *et al.* Sex-biased breeding dispersal is predicted by social environment in birds. *Ecol Evol* **8**, 6483–6491 (2018).
6. Fandos, G. *et al.* Standardised empirical dispersal kernels emphasise the pervasiveness of long-distance dispersal in European birds. *Journal of Animal Ecology* **92**, 158–170 (2023).
7. Jennions, M. D. & Fromhage, L. Not all sex ratios are equal: the Fisher condition, parental care and sexual selection. *Philosophical Transactions of the Royal Society B: Biological Sciences* **372**, 20160312 (2017).

REVIEWERS' COMMENTS

Reviewer #1 (Remarks to the Author):

The authors have adequately addressed the issue I raised considering the limitations of available data. I'm happy with the manuscript as it is now.

We sincerely thank Reviewer #1 for their careful evaluation and constructive feedback throughout the review process. We are grateful for their positive assessment of the revised manuscript.

Reviewer #2 (Remarks to the Author):

In this new version, the authors have demonstrated great effort and dedication. I believe they have responded appropriately and carefully to each of my observations and comments, which is reflected in a much clearer and more precise article.

We sincerely thank Reviewer #2 for their thoughtful and constructive comments, which helped improve the clarity and precision of the manuscript. We greatly appreciate their positive evaluation of the revised version.

Reviewer #3 (Remarks to the Author):

While I appreciate the authors' detailed explanations, I remain unconvinced about the theoretical justification for prioritising the “pre-transformation” approach (Santos/Liker method) over the residual based PGLS approach for Phylogenetic Path Analysis (von Hardenberg and Gonzalez-Voyer, 2013).

We thank the reviewer for the detailed and technically informed comments. We have carefully reconsidered the conceptual issues raised and have revised the manuscript to clarify our rationale and to present both analytical frameworks in the main text.

The authors state that they estimate λ separately for each variable, before computing contrasts. However, as established in the literature for PGLS (e.g., Freckleton et al. 2002; Revell 2010), λ is a parameter of the residual error structure, not the raw traits. Pre-transforming variables independently assumes that the phylogenetic signal in the

response is independent of the predictor, an assumption rarely met in co-evolving biological systems.

We agree that in PGLS, λ is formally estimated as a parameter of the residual covariance structure conditional on the specified response variable. However, in the special case of a single trait model (i.e., no predictors, as in the Santos/Liker method), the trait variance and the residual variance coincide. In that context, λ characterises the phylogenetic signal of the trait itself because there is no additional structured predictor.

It is also worth noting that phylogenetic independent contrasts and phylogenetic generalized least squares are mathematically equivalent formulations of the same Brownian motion model when λ is fixed (Freckleton 2012). Thus, transforming traits prior to analysis represents an alternative parameterisation of the same phylogenetic model rather than a fundamentally different statistical framework.

Thus, whether λ is interpreted as a property of the trait or of the residual covariance structure depends on model specification. The pre-transformation framework follows the logic of univariate comparative methods, in which phylogenetic structure is first accounted for at the trait level prior to evaluating path relationships. In contrast, the residual-based PGLS framework models phylogenetic covariance conditionally within each directed regression.

Furthermore, the authors argue that the von Hardenberg & Gonzalez-Voyer (2013) method is inconsistent because the direction of the regression ($X \rightarrow Y$ vs. $Y \rightarrow X$) yields different λ values. I would argue that this is not a lack of robustness, but a biological requirement of path analysis. Path analysis is a test of directed causal hypotheses; since λ accounts for the unexplained variance in the response, it should change if the hypothesized response variable changes. Forcing symmetry via univariate transformation bypasses this conditional logic leading possibly to inflated type I error rates.

We agree that in residual-based PGLS, λ is inherently conditional on the designated response variable and therefore may differ across alternative causal specifications. This reflects the conditional logic of directed models rather than statistical instability.

However, in systems where multiple traits exhibit substantial phylogenetic signal—as is the case for pre- and post-copulatory mate choice and parental care in our dataset (all $\lambda \approx 0.90–0.98$), whereas ASR and demographic bias show minimal signal ($\lambda \approx 0$)—alternative parameterisations necessarily allocate shared phylogenetic covariance differently.

In the residual-based framework (von Hardenberg & Gonzalez-Voyer method), phylogenetic structure is modelled in the residual variance of the response variable. Thus, when strongly phylogenetically structured traits (e.g., mate choice and parental care) are treated solely as predictors—as in candidate Model 2a (Fig. S2–S3)—their phylogenetic covariance is not explicitly parameterised in the error structure but enters the model through the fixed-effect design matrix. In contrast, when these same traits are specified as responses, the residual variance retains the phylogenetic structure of the trait, resulting in λ estimates approaching unity.

This redistribution of covariance across model structures can, in principle, influence likelihood-based fit statistics (e.g., CIC/AIC) across alternative causal hypotheses. Notably, Model 2a consistently shows substantially lower support under the residual-PGLS implementation (Table S9) relative to the pre-transformation framework (Tables S7–S8), a pattern that likely reflects how phylogenetic covariance is partitioned under different parameterisations rather than instability of inference.

Importantly, we do not interpret this as a methodological flaw. Rather, it highlights that alternative frameworks embody distinct assumptions about how multivariate phylogenetic covariance is distributed across conditional models.

For this reason, we do not rely on a single method. Instead, we implemented both analytical frameworks and present them in parallel in the main text in lines 153-156 & 174-176. Across competing path models, the direction, magnitude and biological interpretation of key paths remain concordant. This convergence indicates that our conclusions regarding the demographic drivers and social consequences of ASR variation are robust to alternative treatments of phylogenetic covariance and are not artefacts of conditional λ estimation.

The use of standard SEM fit indices (CFI, TLI, RMSEA) on λ -transformed PICs remains, in my view, problematic. These indices were developed for independent data sampled from a population. As noted in Freckleton (2009), phylogenetic data, even when transformed, often do not possess the same number of effective degrees of freedom as independent data points ($N-1$). Standard SEM software is 'blind' to the hierarchical source of the contrasts and may therefore produce over-confident fit statistics based on an inflated denominator of independence. To my knowledge, the performance of Global Fit Indices (like CFI and TLI) has not been rigorously validated or even extensively tested on phylogenetically structured contrasts (see Gonzalez-Voyer & von Hardenberg 2014). The 'null' model used to calculate CFI/TLI in lavaan assumes a specific covariance structure that may not be biologically or statistically appropriate when the data origin is a bifurcating tree, even after a λ transformation. Furthermore, even if the contrasts are independent,

they may not be identically distributed if there is heteroscedasticity introduced by the tree topology or if the univariate λ values do not perfectly capture the co-evolutionary signal in the multivariate system.

We thank the reviewer for raising important points regarding the use of global SEM fit indices (CFI, TLI, RMSEA, SRMR) in the context of phylogenetically transformed data.

Phylogenetically independent contrasts (PICs) are statistically independent under a Brownian motion model of trait evolution (Felsenstein 1985; Freckleton 2009), although they are not necessarily identically distributed because contrast variances depend on branch lengths. Given these structural differences between comparative datasets and conventional i.i.d. samples, global fit indices were interpreted cautiously and used descriptively.

We also recognise that indices such as CFI and TLI rely on comparison to a baseline (“null”) model that assumes a particular covariance structure, and that their behaviour has not been extensively evaluated for phylogenetically structured datasets.

In our analyses, model ranking and inference were based primarily on information-theoretic criteria derived from the d-separation framework (Fisher’s C statistic and CICc/AICc), we have clarified this in the Results (line 161) and Methods (lines 539–541), whereas global fit indices were reported descriptively to characterise relative model fit.

Accordingly, our biological conclusions are supported by concordant model rankings and path estimates across alternative analytical frameworks and do not depend on any single global fit statistic.

While it is encouraging that the results are concordant across methods, I strongly suggest the authors move the results obtained with phylopath (which implements the von Hardenberg and Gonzalez-Voyer, 2013 method) to the main text as the primary analysis rather than in the Suppl. Materials. If the authors decide to still keep also the Santos/Liker analysis, I would recommend the inclusion of an acknowledgement that the pre-transformation method, while providing similar results as the PPA approach, is a pragmatic alternative which may however not fully account for co-evolutionary signals in the residuals.

We thank the reviewer for this constructive suggestion. In response, we have revised the manuscript to present the results from both frameworks in the main text (lines 153–156; 174–176), rather than relegating one approach to the Supplementary Materials.

In the revised manuscript, the transformation-based and residual-based approaches are presented in parallel, as complementary implementations that differ in how phylogenetic covariance is parameterised. Importantly, both frameworks yield concordant model rankings and path coefficients.

The pre-transformation approach accounts for phylogenetic structure at the trait level prior to path estimation, whereas the residual-based framework models phylogenetic covariance conditionally within each regression. As noted in the revised Methods (lines 527–531; 552–557), these approaches capture phylogenetic covariance in different ways and therefore rely on somewhat different modelling assumptions.

However, because the biological conclusions are consistent across analytical frameworks, our inference regarding the demographic drivers and social consequences of ASR variation remains robust to differences in modelling framework.

On a minor point, the authors state that they used a “more up-to-date path analysis implementation (piecewiseSEM)”. I would like a clarification on this: PiecewiseSEM implements the exact same d-sep approach as in phylopath, with the only difference that it does not include PGLS models.

We thank the reviewer for this clarification. We have carefully reviewed the manuscript to ensure that no wording suggests that piecewiseSEM represents a “more up-to-date” implementation. We confirm that both piecewiseSEM and phylopath are described as alternative implementations of the same d-separation framework.

Minor comments in the main text:

Line 498 and subsequent: the common definition is “Phylogenetic Path Analysis”, and not “Analyses”

Thank you for this suggestion. We have corrected this to “Phylogenetic Path Analysis” in line 505.

Lines 547-49. You can cite Shipley (2013), which demonstrates the equivalence of AIC (based on the likelihood) with CIC (based on the C statistic).

Thanks for your suggestion, we have cited Shipley (2013) in line 563.

References:

Freckleton, R. P. (2012). Fast likelihood calculations for comparative analyses. *Methods in Ecology and Evolution*, 3(5), 940-947.

Felsenstein, J. (1985). Phylogenies and the comparative method. *The American Naturalist*, 125(1), 1-15.

Freckleton, R. P. (2009). The seven deadly sins of comparative analysis. *Journal of evolutionary biology*, 22(7), 1367-1375.